# Nuclear condensates of p300 formed though the structured catalytic core can act as a storage pool of p300 with reduced HAT activity

Yi Zhang [1], Kyle Brown[2], Yucong Yu[3], Ziad Ibrahim[4], Mohamad Zandian [1], Hongwen Xuan[3], Steven Ingersoll[2], Thomas Lee[5], Christopher C. Ebmeier [5], Jiuyang Liu[1], Daniel Panne[4], Xiaobing Shi [3], Xiaojun Ren[2] & Tatiana G. Kutateladze [1✉]

The transcriptional co-activator and acetyltransferase p300 is required for fundamental cellular processes, including differentiation and growth. Here, we report that p300 forms phase separated condensates in the cell nucleus. The phase separation ability of p300 is regulated by autoacetylation and relies on its catalytic core components, including the histone acetyltransferase (HAT) domain, the autoinhibition loop, and bromodomain. p300 condensates sequester chromatin components, such as histone H3 tail and DNA, and are amplified through binding of p300 to the nucleosome. The catalytic HAT activity of p300 is decreased due to occlusion of the active site in the phase separated droplets, a large portion of which co-localizes with chromatin regions enriched in H3K27me3. Our findings suggest a model in which p300 condensates can act as a storage pool of the protein with reduced HAT activity, allowing p300 to be compartmentalized and concentrated at poised or repressed chromatin regions.

[1] Department of Pharmacology, University of Colorado School of Medicine, Aurora, CO, USA. [2] Department of Chemistry, University of Colorado, Denver, CO, USA. [3] Center for Epigenetics, Van Andel Research Institute, Grand Rapids, MI, USA. [4] Department of Molecular and Cell Biology, Leicester Institute of Structural and Chemical Biology, University of Leicester, Leicester, UK. [5] Department of Biochemistry, University of Colorado, Boulder, CO, USA. ✉email: tatiana.kutateladze@cuanschutz.edu

F ormation of condensates in the cell nucleus has been shown to drive the assembly of nuclear bodies, Cajal bodies, nucleoli, and speckles[1,2]. These membraneless compartments are produced via the liquid–liquid phase separation (LLPS) mechanisms and depend on weak, multivalent interactions involving intrinsically disordered regions (IDRs) of biomolecules[3–5]. Several nuclear proteins and chromatin regulators, including HP1, BRD4, CBX, and MORC3, have been shown to form condensates with their phase separation ability being necessary for biological functions and subnuclear localization of these proteins[6–16].

The transcriptional co-activator p300 is an acetyltransferase that is frequently dysregulated in disease, particularly cancer. p300 associates with over 400 binding partners, including transcription factors, DNA-binding proteins and subunits of the RNA polymerase II complex, to form activation complexes and facilitate transcription[17–21]. p300 acetylates histones and a number of essential nonhistone proteins, notably p53, E2F1, and androgen and estrogen receptors[22–24]. The highly selective acetylation of the H3K27 and H3K18 sites by p300 requires a cooperative action of its two functional domains: acetyllysine-binding bromodomain (BD) and the H3 tail-binding ZZ domain[25].

p300 and its close homolog CBP share the same domain architecture, consisting of several conserved modules, including the catalytic histone acetyltransferase (HAT) domain. Located in the middle of the protein, the HAT domain is surrounded by BD, a RING finger, and a plant homeodomain finger (PHD) from one side and the ZZ domain from another side that together comprise the catalytic core of p300 (Fig. 1a). The p300 catalytic activity is regulated through acetylation/deacetylation of a long loop in the HAT domain, termed the autoinhibitory loop (AIL), which in a hypoacetylated form impedes the acetyltransferase activity of p300 but releases the inhibition upon hyper autoacetylation[17,26]. Structural studies of the p300 region encompassing BD, RING, PHD and the HAT domain suggest an additional layer of p300 regulation through the RING finger that contributes to the inhibition by occluding the HAT active site[27]. Furthermore, the BD–RING–PHD–HAT region of p300 has been shown to form a dimer, in which the hypoacetylated form of AIL of one molecule inserts in the active site of the HAT domain from another molecule and thus undergoes 'in trans' autoacetylation, a reaction that is controlled by signal-dependent transcription factor dimerization[28]. Genome-wide-mapping analysis reveals that p300/CBP generally occupies transcriptionally active chromatin marked by the active H3K27ac modification, however a large number of silent or poised genes marked by the repressive H3K27me3 modification and bound by p300/CBP has also been identified[25,29–33]. In cells, p300/CBP is ubiquitously expressed and can form discrete foci in the nucleus and localize to PML nuclear bodies[34–37].

In this study, we show that the structured catalytic core of p300 undergoes phase separation and forms liquid condensates in cells and in vitro. The acetylation state of p300 modulates its ability to phase separate through attenuating intermolecular interactions involving the autoinhibition loop, the HAT domain, and BD. Our data suggest a model for compartmentalization and concentration of a pool of p300 with reduced catalytic activity at poised or repressed chromatin regions through the formation of phase-separated condensates.

## Results and discussion

### p300 forms dynamic condensates in cells

p300 is highly abundant in the cell nucleus and can shuttle between the nucleoplasmic and cytoplasmic fractions[38–40]. To characterize the distribution of p300, we transfected HeLa cells with YFP-tagged full-length p300 (YFP-p300$_{FL}$) and monitored the localization of YFP-p300$_{FL}$ in live cells by fluorescence microscopy (Fig. 1b–d). In the majority of cells assayed, YFP-p300$_{FL}$ was somewhat diffusely distributed throughout the nucleus, but 15–30% of cells displayed nuclear speckles various in size (Fig. 1b). These data indicate the presence of discrete YFP-p300$_{FL}$ compartments with elevated protein concentration and are in line with previous reports showing that both endogenous and expressed p300 (and its homolog CBP) form dynamic nuclear bodies[34–37].

To determine whether p300 is mobile in the speckles, we measured the diffusion kinetics of selected p300 puncta by fluorescence recovery after photobleaching (FRAP) experiments. For each selected region, the laser beam was applied after two initial scans, and cell images were collected at 10 s intervals for a duration of 280 s (Fig. 1c, d). The experiments were performed on seven cells, and changes in the fluorescence signal of each bleached region were analyzed. The averaged signal intensity, which was normalized and plotted (Fig. 1c), showed a ~60% recovery of the bleached regions' fluorescence with a half-life ($t_{1/2}$) of 59 s. Because the photobleaching irreversibly abolishes fluorescence, the recovery of the signal implies that YFP-p300$_{FL}$ readily moves from unbleached regions to the bleached region. The fast signal recovery, and therefore the fast diffusion of p300, suggest that YFP-p300$_{FL}$ speckles are viscous liquid droplets characterized by rapid protein exchange with the surrounding environment. Collectively, these results demonstrate that full length p300 forms dynamic nuclear condensates with liquid/gel-like properties.

### The catalytic core of p300 phase separates into liquid droplets in cells and in vitro

The middle part of p300, consisting of the BD followed by the RING, PHD, HAT, and ZZ domains, comprises the catalytic core that we postulated might be involved in the phase separation process. To test this idea, we generated YFP-tagged truncated p300 construct (aa 1024–1830 of p300), which in addition to the catalytic core contains a TAZ2 domain and is referred to as p300$_{BRPHZT}$. We have previously shown that p300$_{BRPHZT}$ associates with chromatin in vitro and in vivo comparably to full-length p300[25]. The YFP-p300$_{BRPHZT}$ protein was produced using a doxycycline (DOX) inducible pTripZ vector, and expression of p300$_{BRPHZT}$ was visualized in live HeLa cells. Much like the full-length protein, YFP-p300$_{BRPHZT}$ localized primarily to the nucleus and formed foci of various sizes (Supplementary Fig. 1). Furthermore, FRAP experiments showed a ~80% fluorescence recovery with $t_{1/2} = 29$ s, indicating that YFP-p300$_{BRPHZT}$ and full length YFP-p300 have similar capabilities to form dynamic condensates in cells (Fig. 1e).

The cell nucleus is a highly crowded and viscous compartment packed with DNA, histones, and a multitude of nuclear proteins. To mimic molecular crowding, polyethylene glycol (PEG) is frequently used in vitro. Addition of PEG (PEG3350) to a solution containing the purified catalytic core, p300$_{BRPHZ}$ (aa 1035–1720 of p300), rapidly induced cloudiness in the originally clear solution, indicative of the conversion to a heterogenous suspension (described in detail below). Under a microscope, we observed spherical droplets of various sizes, suggesting the formation of liquid–liquid phase-separated condensates of the catalytic core of p300 (Fig. 1f). About 35% of p300$_{BRPHZ}$ remained in the supernatant after removing the droplets by centrifugation (Supplementary Fig. 2).

### Self-acetylation of p300 impairs its ability to phase separate

We found that different batches of purified p300$_{BRPHZ}$ showed variable ability to form droplets under otherwise identical conditions (Figs. 1f and 2a, b). Recombinantly expressed p300 is auto-acetylated to varying degrees[26,41]. We, therefore, assessed

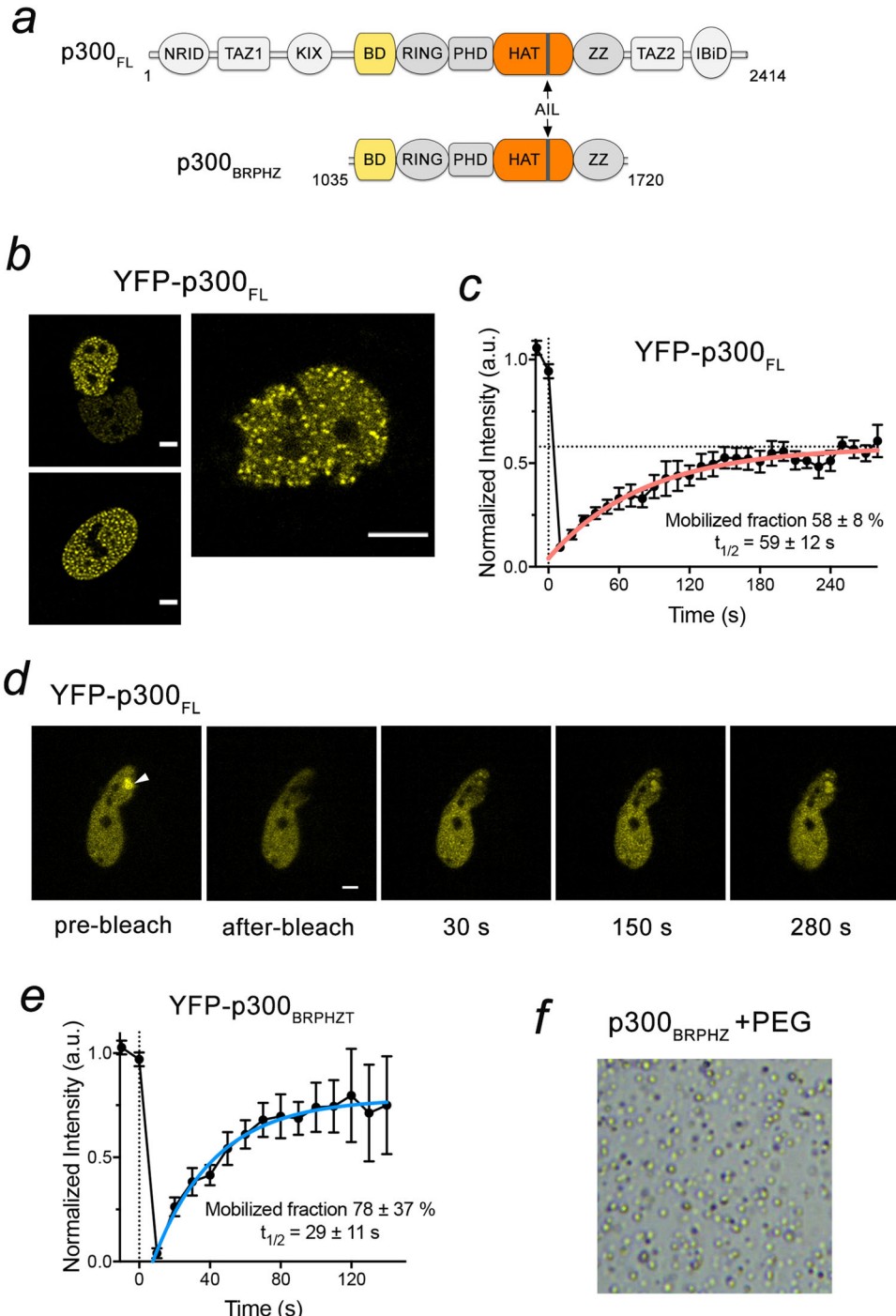

**Fig. 1 p300 phase separates to condensates in living cells. a** Schematic of p300$_{FL}$ and p300$_{BRPHZ}$. **b** Representative images of HeLa cells expressing YFP-p300$_{FL}$. Scale bar, 5 μm. **c** FRAP curve of YFP-p300$_{FL}$ was obtained from averaging data from six HeLa cells. Error bars represent SEM. $n = 6$ cells examined over three independent experiments. **d** Representative FRAP images of YFP-p300$_{FL}$ expressed in HeLa cells. The images were taken before and after photobleaching at indicated time points. The bleached condensate is indicated by a white arrowhead. Scale bar, 5 μm. **e** FRAP curve of YFP-p300$_{BRPHZT}$ condensates in HeLa cells. The FRAP curve was obtained from averaging data from four cells. Error bars represent SEM. **f** A representative image of a sample (from at least three replicates) containing 13 μM p300$_{BRPHZ}$ and 12% PEG3350 (PEG) in a 100 μm × 100 μm square region under a microscope.

the self-acetylation levels of two different batches of purified p300$_{BRPHZ}$ by liquid chromatography–mass spectrometry (LC–MS). Acetylation levels of p300$_{BRPHZ}$ were highly heterogeneous, containing a series of species with discrete molecular mass. For example, purification 1 contained p300$_{BRPHZ}$ with molecular mass ranging from ~81 to 81.7 kDa (Fig. 2a, c, and d). The increment of each mass peak was ~42 Da, the mass of one

acetyl group. Compared with its theoretical mass of 80,548.5 Da, recombinantly expressed p300$_{BRPHZ}$ from purification 1 was acetylated on 13–25 lysine residues, with a median of 16 acetylated lysines (Fig. 2c, d). The sequence of p300$_{BRPHZ}$ contains 60 lysine residues, indicating that ~20–40% of lysine residues were acetylated. The p300$_{BRPHZ}$ protein from purification 2 had on average 26 acetylated lysines. p300$_{BRPHZ}$ with the lower

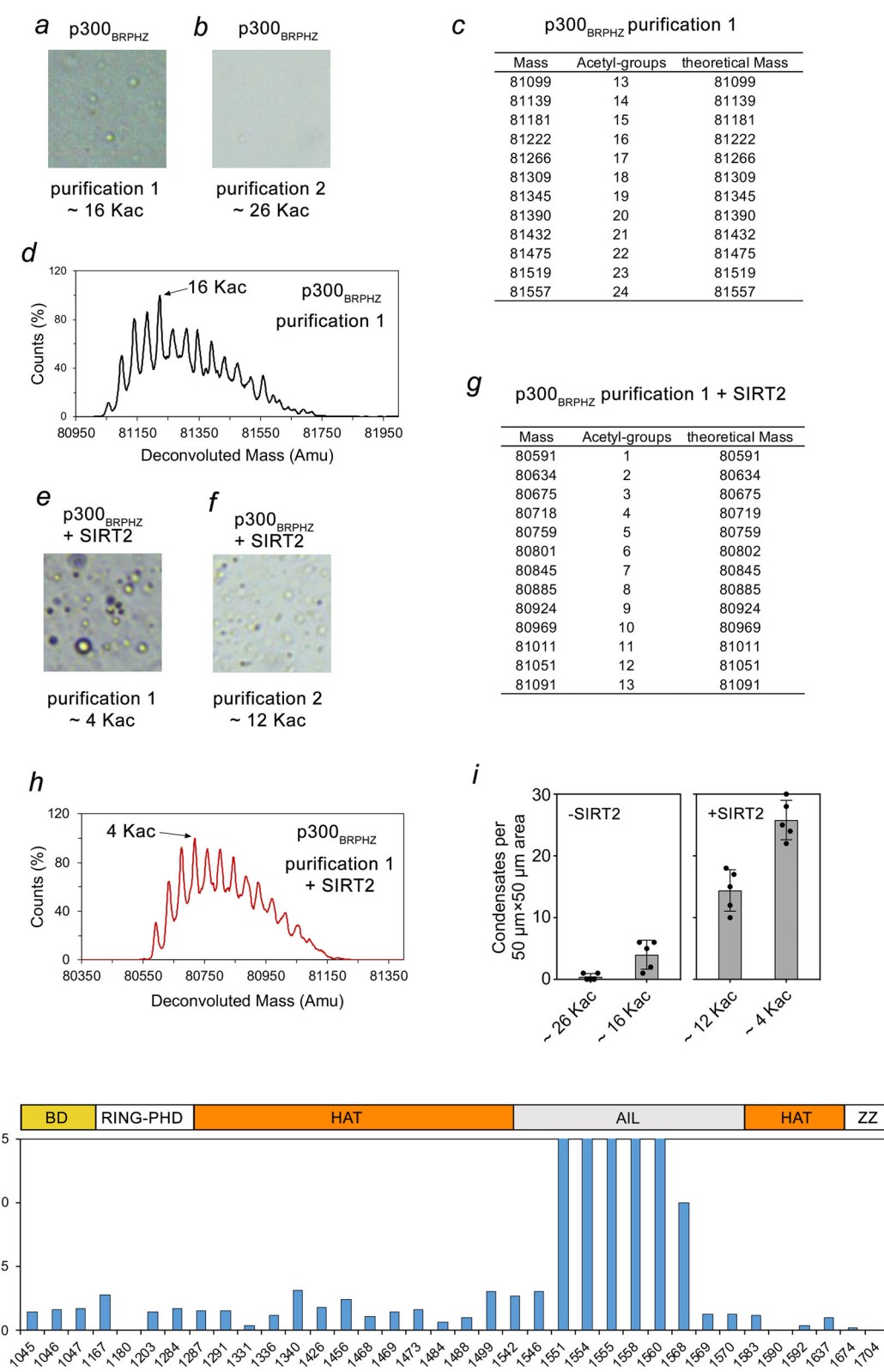

**c** p300$_{BRPHZ}$ purification 1

| Mass | Acetyl-groups | theoretical Mass |
|---|---|---|
| 81099 | 13 | 81099 |
| 81139 | 14 | 81139 |
| 81181 | 15 | 81181 |
| 81222 | 16 | 81222 |
| 81266 | 17 | 81266 |
| 81309 | 18 | 81309 |
| 81345 | 19 | 81345 |
| 81390 | 20 | 81390 |
| 81432 | 21 | 81432 |
| 81475 | 22 | 81475 |
| 81519 | 23 | 81519 |
| 81557 | 24 | 81557 |

**g** p300$_{BRPHZ}$ purification 1 + SIRT2

| Mass | Acetyl-groups | theoretical Mass |
|---|---|---|
| 80591 | 1 | 80591 |
| 80634 | 2 | 80634 |
| 80675 | 3 | 80675 |
| 80718 | 4 | 80719 |
| 80759 | 5 | 80759 |
| 80801 | 6 | 80802 |
| 80845 | 7 | 80845 |
| 80885 | 8 | 80885 |
| 80924 | 9 | 80924 |
| 80969 | 10 | 80969 |
| 81011 | 11 | 81011 |
| 81051 | 12 | 81051 |
| 81091 | 13 | 81091 |

acetylation level (purification 1) visibly formed condensates in the presence of PEG (Fig. 2a), while p300$_{BRPHZ}$ with the higher acetylation level (purification 2) showed almost no droplet formation (Fig. 2b). Together, these results suggest a negative correlation between the acetylation level of p300 and its ability to form condensates in vitro.

We further examined whether acetylation levels modulate the condensation behavior of p300$_{BRPHZ}$. The NAD$^+$-dependent histone deacetylase SIRT2 has been shown to selectively and uniquely deacetylate p300 in cells and in vitro[42]. We incubated p300$_{BRPHZ}$ with SIRT2 and NAD$^+$ and then purified the protein by size-exclusion chromatography. This reduced the number of

**Fig. 2 p300$_{BRPHZ}$ is heterogeneously acetylated and SIRT2-induced deacetylation promotes phase separation of p300$_{BRPHZ}$. a, b** Representative images of two independently expressed and purified p300$_{BRPHZ}$ samples containing 13 μM protein and 12% PEG in a 50 μm × 50 μm square region on cover slides under a microscope. **c, d** Liquid chromatography–mass spectrometry (LC–MS) analysis of p300$_{BRPHZ}$ shown in (**a**). A table of peak mass and the corresponding number of acetylated lysine residues identified by mass spectrometry analysis are shown in (**c**) and (**d**). **e, f** Representative images of p300$_{BRPHZ}$ samples shown in (**a**) and (**b**) were treated with the deacetylase SIRT2. **g, h** LC–MS analysis of p300$_{BRPHZ}$ shown in (**e**). A table of peak mass and the corresponding number of acetylated lysine residues identified by mass spectrometry analysis are shown in (**g**) and (**h**). **i** Quantification of droplets counted in images acquired for the samples in (**a, b, e, f**). The number of droplets was counted in five non-overlapping 50 μm × 50 μm square regions and the mean value was plotted. Error bars represent SD. **j** Acetylated sites in p300$_{BRPHZ}$ identified by LC–MS/MS. Vertical bars indicate differences in signal intensities of identical tryptic peptides derived from untreated vs. SIRT2-treated samples on the log$_2$ scale. The highest intensities exceeding the plot were essentially qualitative, as they were only present in the untreated sample. A schematic representation of p300$_{BRPHZ}$ is shown above. Source data are provided in the Source Data file.

acetylated lysines in p300$_{BRPHZ}$ (purification 1) from a median of 16 to a median of 4 (Fig. 2c, d, g, h). The same treatment of p300$_{BRPHZ}$ (purification 2) reduced the acetylation level from a median of 26 to 12 acetylated lysines. Generally, lower acetylation levels were correlated with an increased propensity to form droplets (Fig. 2i).

To identify regions in p300 affected by SIRT2 treatment, we analyzed tryptic peptides derived from untreated and SIRT2-treated p300$_{BRPHZ}$ using liquid chromatography–tandem mass spectrometry (LC–MS/MS). 49 out of 60 lysine residues were acetylated in untreated p300$_{BRPHZ}$. Comparison of the signal intensities of peptides revealed substantial differences in acetylation levels of untreated and SIRT2-treated p300$_{BRPHZ}$ (Fig. 2j). The most notable changes were observed for K1551, K1554, K1555, K1557, and K1560 located in the AIL, as acetylation of these lysine residues was not detectable in the SIRT2-treated p300$_{BRPHZ}$ sample. Another AIL residue, K1568, also displayed a considerable reduction in signal intensity upon SIRT2 treatment. In contrast, signal intensity for lysine residues located in other regions of p300$_{BRPHZ}$ was reduced to a substantially lesser degree. These data corroborate the idea that p300 self-acetylates a wide range of lysine residues spanning the catalytic BD–RING–PHD–HAT–ZZ core, and that the auto-inhibitory loop undergoes fast deacetylation, whereas deacetylation of other regions of p300$_{BRPHZ}$ occurs much slower.

Because our data suggest that acetylation of p300 negatively regulates its ability to phase separate, we reasoned that acetylation should decrease or even eliminate the PEG-induced p300$_{BRPHZ}$ droplet formation. To test this, we first measured the HAT activity of untreated and SIRT2-treated p300$_{BRPHZ}$ in the presence of acetyl-CoA by monitoring the release of the CoA product over time. As expected, SIRT2-treated p300$_{BRPHZ}$ produced more CoA than untreated p300$_{BRPHZ}$, due to the higher number of unmodified lysine residues (substrates) present in SIRT2-treated p300$_{BRPHZ}$ (Fig. 3a). The acetylation reaction was fast and completed before 20 min, in keeping with the previously reported activity of the p300 HAT domain[41]. Adding acetyl-CoA to a SIRT2-treated p300$_{BRPHZ}$/PEG suspension led to a decrease in the cloudiness of the sample within 1 min (Fig. 3b), and liquid droplets were no longer visible under the microscope (Fig. 3c, d, and Supplementary Fig. 3a), indicating a disruption of LLPS due to the HAT reaction. Collectively, these data suggest that auto-acetylation of the AIL decreases the formation of p300$_{BRPHZ}$ condensates.

**Phase separation of p300 relies on both BD and AIL.** Weak and often nonspecific multivalent interactions are believed to drive phase separation[1,4,7,8]. The p300 HAT domain has been reported to associate with the hypoacetylated AIL from another p300 molecule[28] and BD of CBP was shown to bind the AIL peptide acetylated at K1596[43]. Accordingly, we envisage two distinct mechanisms for the transition of the hyperacetylated p300

catalytic core (incapable of phase separation) to the hypoacetylated p300 catalytic core (capable of phase separation), which rely on intermolecular 'in trans' HAT-AIL or BD–acetyl–lysine (Kac) interactions (Fig. 3e). To test these possibilities, we examined the role of the BD and AIL in the formation of condensates. We used a deletion of the entire autoinhibitory loop (ΔAIL) in p300$_{BRPHZ}$ or a mutation N1132A, which was previously shown to abrogate acetyllysine binding of the BD[27].

Compared to the WT p300$_{BRPHZ}$ protein that contained on average ~12 Kac sites and readily underwent phase separation (Fig. 3f), the N1132A mutant did not form PEG-induced droplets, despite having a similar acetylation level (~14 Kac sites) (Fig. 3f–h). Likewise, the ΔAIL p300$_{BRPHZ}$ mutant with ~7 Kac sites formed less visible condensates than the WT protein with ~4 Kac sites (Fig. 3i–k). We note that although the entire AIL was deleted, the ΔAIL p300$_{BRPHZ}$ mutant was still acetylated at 7 lysine residues located outside this loop in the BD–RING–PHD–HAT–ZZ region. These data suggest that both HAT–AIL and BD–Kac-binding mechanisms contribute to p300$_{BRPHZ}$ phase separation. In a hyper-acetylated form where both the AIL and other regions of p300$_{BRPHZ}$ are acetylated, BD may favor intramolecular 'in cis' contacts with the acetylated AIL, limiting the probability of intermolecular interactions and leading to a diffused distribution of p300$_{BRPHZ}$. Upon SIRT2-treatment of WT p300$_{BRPHZ}$, deacetylation of the AIL results in the release of BD, allowing intermolecular 'in trans' BD–Kac interactions and both intra- and intermolecular HAT–AIL interactions (Fig. 3e).

In support of this model, SIRT2 treatment of N1132A p300$_{BRPHZ}$, which did not originally phase separate (Fig. 4a, left panel), led to the condensate formation (Fig. 4a, right panel, and Fig. 4b), suggesting that deacetylation of AIL promotes phase separation through the HAT–AIL interaction (Fig. 4c). Furthermore, both SIRT2-treated or untreated ΔAIL p300$_{BRPHZ}$ formed droplets, although to a lesser degree compared to the WT protein, reinforcing the role of the BD–Kac interaction in promoting phase separation (Fig. 4d–f). Moreover, the catalytically impaired mutant D1399A p300$_{BRPHZ}$ showed the formation of the droplets through the HAT–AIL interaction, since no self-acetylation occurred in this mutant (Fig. 4g, h, and Supplementary Fig. 2b, c). The SIRT2-dependent phase separation was also observed in the p300$_{HZ}$ construct containing only the HAT and ZZ domains (Fig. 4i, j), whereas further deletion of AIL completely abolished droplet formation, regardless of the SIRT2 treatment (Fig. 4k, l). Lastly, both N1132A and ΔAIL mutants of YFP-p300$_{FL}$ formed droplets in HeLa cells, confirming that full-length p300 can phase separate through either HAT–AIL interaction in N1132A YFP–p300$_{FL}$ or BD–acetyl–lysine interaction in ΔAIL YFP–p300$_{FL}$ (Fig. 4m).

To further confirm our model, we performed small-angle X-ray scattering (SAXS) experiments on purified WT p300$_{340–2094}$ and ΔAIL p300$_{340–2094}$ proteins at various time points before and after incubating with acetyl-CoA (Fig. 5a). SAXS experiments provide

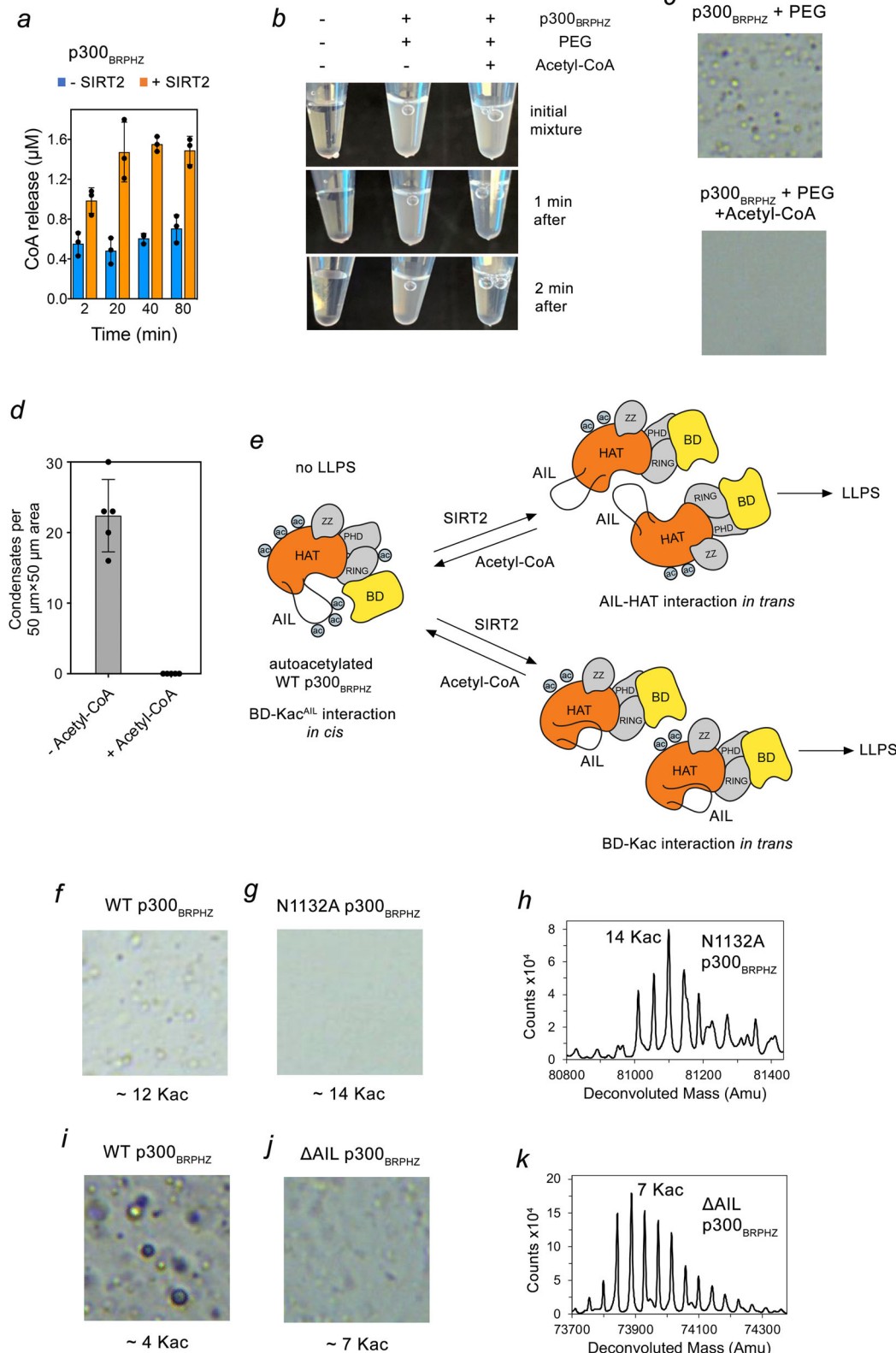

information about the protein's size and shape in solution. For WT p300$_{340-2094}$, autoacetylation led to little changes in the SAXS profile: the Rg value remained unchanged at ~10 nm, in agreement with monomeric p300, possibly because the acetylated AIL engages the BD 'in cis' (Fig. 5b, top panel). In contrast, for ΔAIL p300$_{340-2094}$, the Rg value increased over time (to ~35 nm) after autoacetylation for 1.5 h resulting in spherical droplets with

a maximum dimension of ~125 nm (Fig. 5a, bottom panel). These droplets likely arise through 'in trans' engagement of the BD with acetylated lysine residues outside AIL (Fig. 5b, bottom panel).

**p300 condensates sequester nucleosomal substrates.** Besides auto-acetylation, p300 catalyzes acetylation of lysine residues of

**Fig. 3 p300$_{BRPHZ}$ phase separation requires both AIL and BD. a** HAT activity of untreated p300$_{BRPHZ}$ (blue) and SIRT2-treated p300$_{BRPHZ}$ (orange) measured by a fluorometric assay. Reactions were started by the addition of acetyl-CoA and quenched by flash-freeze at indicated time points. Data are presented as mean values ± SD; error bars represent SD from triplicate measurements. **b** Phase separation of SIRT2-treated p300$_{BRPHZ}$. The reaction mixture contained 10 μM p300$_{BRPHZ}$ and 12% PEG. Addition of Acetyl-CoA led to the disassembly of the p300$_{BRPHZ}$ droplets. **c** Representative images of p300$_{BRPHZ}$ samples from (*k*) after incubation for 2 min under a microscope. **d** Quantification of droplets counted in images acquired for the sample in (**c**). The number of droplets was counted in five non-overlapping 50 μm × 50 μm square regions and the mean value was plotted. Error bar represents SD. **e** Schematics of the p300$_{BRPHZ}$ phase separation mechanisms that can occur simultaneously. Multivalent '*in trans*' interactions between HAT and deacetylated AIL and/or BD and acetylated lysines outside AIL can promote the formation of condensates. **f, g** Representative images of WT p300$_{BRPHZ}$ (**f**) and N1132A mutant (**g**) samples (from at least three replicates) containing 13 μM protein and 12% PEG in a 50 μm × 50 μm square region on cover slides under a microscope. **h** LC–MS analysis of N1132A p300$_{BRPHZ}$ shown in (**g**). **i, j** Representative images of WT p300$_{BRPHZ}$ (**i**) and ΔAIL mutant (**j**) samples (from at least three replicates) containing 13 μM protein and 12% PEG in a 50 μm × 50 μm square region on cover slides under a microscope. **k** LC–MS analysis of ΔAIL p300$_{BRPHZ}$ shown in (**j**). Source data are provided in the Source Data file.

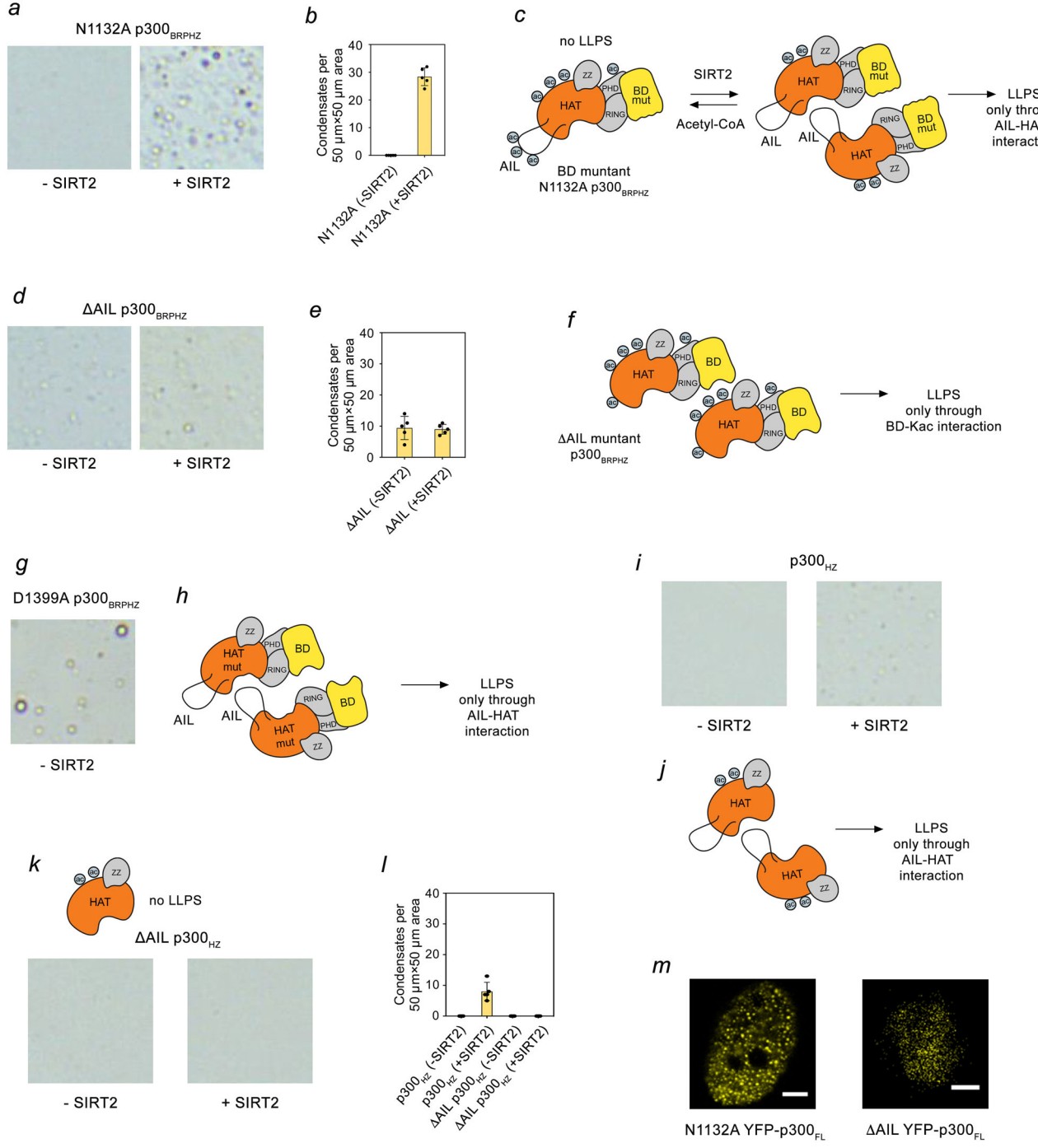

**Fig. 4 The molecular mechanisms of p300_BRPHZ phase separation. a** Representative images of untreated or SIRT2-deacetylated N1132A p300_BRPHZ samples containing 13 μM protein and 12% PEG in a 50 μm × 50 μm square region on cover slides under the microscope. **b** Quantification of droplets counted in images acquired for the sample in (**a**). The number of droplets was counted in five non-overlapping 50 μm × 50 μm square regions and the mean value was plotted. Error bar represents SD. **c** Schematic of the N1132A p300_BRPHZ phase separation via the intermolecular interaction between the HAT domain and deacetylated AIL. **d** Representative images of untreated or SIRT2-deacetylated ΔAIL p300_BRPHZ samples containing 13 μM protein and 12% PEG in a 50 μm × 50 μm square region on cover slides under a microscope. **e** Quantification of droplets counted in images acquired for the sample in (**d**). The number of droplets was counted in five non-overlapping 50 μm × 50 μm square regions and the mean value was plotted. Error bar represents SD. **f** Schematic of the ΔAIL p300_BRPHZ phase separation via the intermolecular interaction between BD and acetylated lysines outside AIL. **g** A representative image of purified D1399A p300_BRPHZ sample (from at least three replicates) containing 13 μM protein and 12% PEG in a 50 μm × 50 μm square region on cover slides under the microscope. **h** Schematic of the catalytically inactive D1399A p300_BRPHZ phase separation via the intermolecular interaction between the HAT domain and unmodified AIL. **i** Representative images of untreated or SIRT2-deacetylated WT p300_HZ samples containing 13 μM protein and 12% PEG in a 50 μm × 50 μm square region on cover slides under the microscope. **j** Schematic of the p300_HZ phase separation via the intermolecular interaction between the HAT domain and deacetylated AIL. **k** Representative images of untreated or SIRT2-deacetylated ΔAIL p300_HZ samples containing 13 μM protein and 12% PEG in a 50 μm × 50 μm square region on cover slides under the microscope. **l** Quantification of droplets counted in images acquired for the samples in (**i**, **k**). The number of droplets was counted in five non-overlapping 50 μm × 50 μm square regions and the mean value was plotted. Data are presented as mean values ± SD; the error bar represents SD. **m** Representative images of HeLa cells (from at least three replicates) expressing N1132A and ΔAIL mutants of YFP-p300_FL. Scale bar, 5 μm.

histone proteins in nucleosomes, particularly at H3K18 and H3K27 sites[22,23,25]. We, therefore, tested whether p300 droplets could concentrate nucleosomal components, such as DNA and histone tails. We monitored the recruitment of fluorescein (FAM)-labeled histone H3 tail (FAM-H3, residues 1–12 of H3) and FAM-labeled 37 bp double-stranded DNA (FAM-DNA) to p300_BRPHZ droplets using confocal microscopy (Fig. 6a and Supplementary Fig. 3b). The p300 condensates exhibited bright fluorescence when incubated with FAM-DNA or FAM-H3, but not when incubated with the control FAM. The recruitment of histone H3 tail to condensates was due to binding of the ZZ domain of p300 to the unmodified H3 tail[25], however, p300 and CBP were thought to not contact DNA themselves[34]. We noticed lower background fluorescence when p300 condensates were incubated with FAM-DNA compared to FAM-H3, which suggests that p300_BRPHZ concentrates better and therefore binds stronger to DNA than to the histone H3_{1-12} peptide. Overall, these results demonstrate that p300_BRPHZ condensates can sequester nucleosomal components like histone H3 tail and DNA from the surrounding environment.

Can p300 bind DNA and is this reaction mediated by p300 self-acetylation? We tested the association of untreated and SIRT2-treated p300_BRPHZ with 601 DNA (147 bp) by electrophoretic mobility shift assay (EMSA). The SIRT2-treated p300_BRPHZ, containing on average 4 Kac outside the AIL, strongly bound to 601 DNA with an apparent $K_d$ of ~0.3 μM (Fig. 6b). In contrast, untreated p300_BRPHZ, which contains on average ~16 Kac including in the AIL, showed much weaker binding to 601 DNA in the same condition (Fig. 6c). DNA binding of p300_BRPHZ, therefore, is directly regulated by its acetylation level, as acetylation neutralizes the positive charge of the unmodified lysine side chain, leading to a decrease in binding to the negatively charged DNA. We also found that auto-acetylation regulates p300_BRPHZ association with nucleosomes. While SIRT2-treated p300_BRPHZ bound to the reconstituted nucleosome core particle (NCP) with a low micromolar $K_d$ in EMSA assays (Fig. 6d), the association of untreated p300_BRPHZ with NCP was notably compromised (Fig. 6e). Collectively, these data suggest that the phase transition and nucleosome binding functions of p300_BRPHZ are linked to and regulated by its auto-acetylation.

The results described above may suggest that DNA and the HAT domain compete for the same hypoacetylated AIL, which could lead to a decrease in phase separation ability of p300_BRPHZ. However, the addition of an equimolar amount of NCP to p300_BRPHZ stimulated the formation of more and larger droplets, whereas NCPs alone did not phase separate under the same

condition (Fig. 6f). These data imply that even when the AIL binds to DNA and is, therefore, less available to contribute to the droplet formation through the HAT–AIL mechanism, the intramolecular association through the BD–Kac mechanism is sufficient to maintain phase separation.

**Catalytic HAT activity of p300 is decreased in condensates.** We examined the effect of phase separation on enzymatic activity and histone substrate selectivity of p300_BRPHZ. Because phase separation conditions can promote enzymatic reactions owing to higher local concentrations of both the enzyme and substrate[44], we initially thought that p300 HAT activity would also be elevated due to increased local concentration of p300_BRPHZ and the substrate, either p300_BRPHZ and/or NCP inside the droplets. On the other hand, the intermolecular AIL–HAT interaction, which leads to the formation of p300 condensates, physically blocks the active site of the HAT domain and thus would lead to a reduction in the enzymatic activity, similar to what was observed for the TOR kinase[45]. In addition to sterically occluding the substrate–enzyme complex formation, the crowded viscous environment in condensates slows diffusion of substrates and products, which results in the reduced catalytic activity of enzymes[46,47].

To compare the HAT activity in solution and phase-separated droplets, we first monitored the self-acetylation of p300_BRPHZ. As shown in Fig. 6g, the addition of PEG to the SIRT2-treated p300_BRPHZ, which does not phase separate because of low protein concentration (0.2 μM), had essentially no effect on the change in the fluorescence signal (due to CoA release) over time, suggesting that the presence of PEG did not alter the catalytic activity of p300_BRPHZ (Fig. 6g). In contrast, the change in fluorescence was slower in the PEG-induced phase-separated suspension (Fig. 6h, red line) or in the supernatant after the droplets were spun down (Fig. 6i) compared to the change in fluorescence in the solution without droplets (Fig. 6h, blue line). These data indicate that autoacetylation is inhibited when p300_BRPHZ forms condensates.

We next measured the acetyltransferase activity of p300_BRPHZ on reconstituted nucleosomes in homogenous solutions and under the phase separation condition. SIRT2-treated (containing ~12 Kac) p300_BRPHZ and untreated (containing ~26 Kac) p300_BRPHZ were mixed with NCP at a 1:1 ratio, and after the addition of acetyl-CoA, the CoA release was monitored at indicated time points. The reactions with NCP as a substrate (Fig. 6j, blue and orange bars, and Supplementary Fig. 4a) were completed within 2 min for both SIRT2-treated and untreated p300_BRPHZ and were much faster than p300_BRPHZ autoacetylation

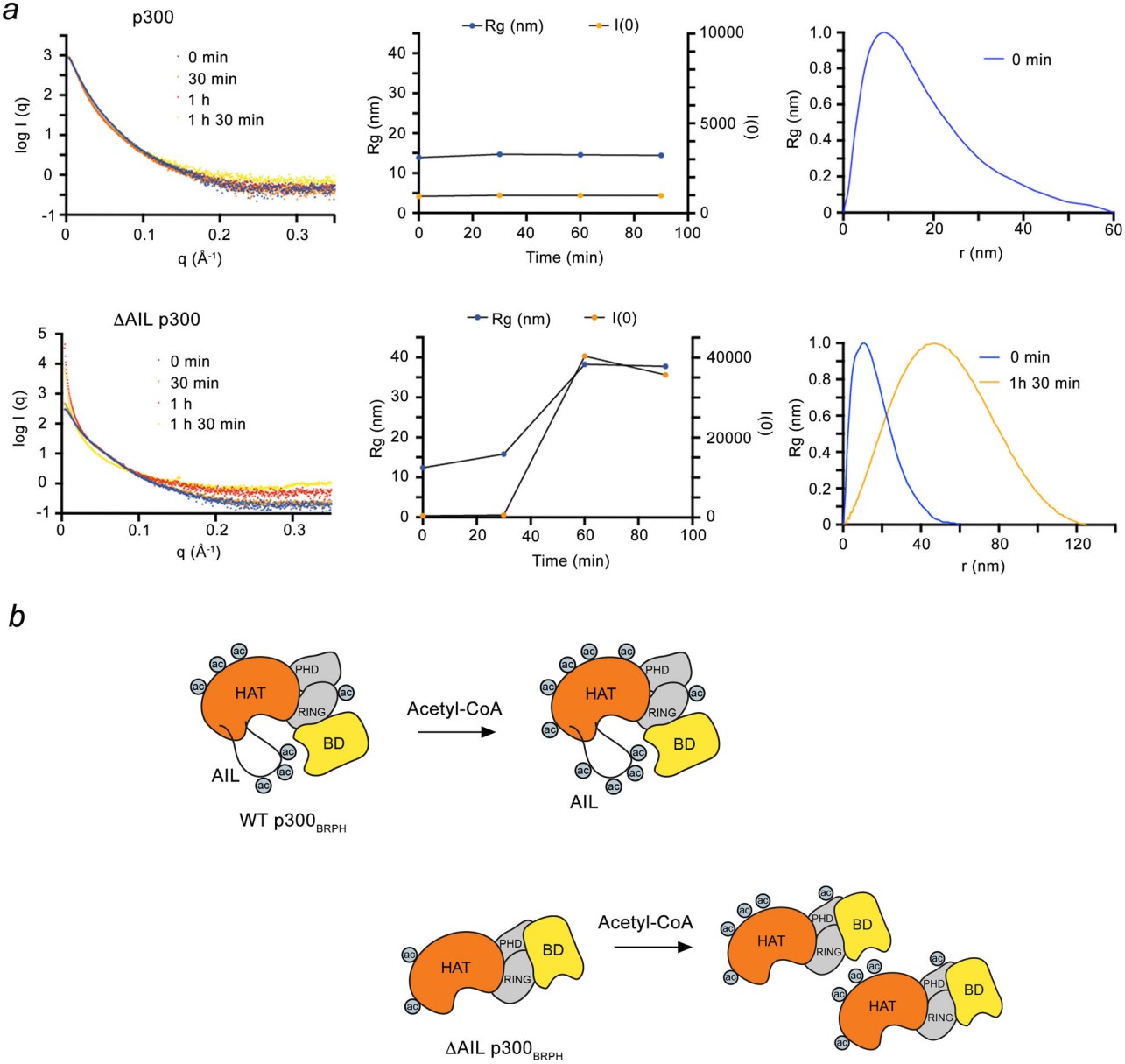

**Fig. 5 ΔAIL p300₍₃₄₀₋₂₀₉₄₎ interacts 'in trans' in solution. a** Small angle X-ray scattering (SAXS) data collected for WT (upper panel) and ΔAIL (lower panel) p300₍₃₄₀₋₂₀₉₄₎ upon addition of acetyl-CoA at indicated time points including 0 min (blue) and 90 min (orange). SAXS intensity (left), Rg plots (middle), and P(r) functions (right) determined from the experimental scattering data were shown. **b** Schematics of the WT and ΔAIL p300₍ᵦᵣₚₕ₎ intramolecular and intermolecular interactions between BD and acetylated lysines.

(Fig. 3a). The HAT reaction was also carried out in a suspension of PEG-induced droplets of SIRT2-treated p300₍ᵦᵣₚₕz₎ and NCP. Again, as in the case of autoacetylation (Supplementary Fig. 4b, gray bars), the apparent HAT activity of p300₍ᵦᵣₚₕz₎ in the phase-separated condensates was decreased compared to the HAT activity of p300₍ᵦᵣₚₕz₎ in solution (Fig. 6j, gray bars).

To assess the histone substrate selectivity of p300₍ᵦᵣₚₕz₎, we monitored acetylation of NCP by western blot using antibodies against H3K27ac, H3K9ac, and H3K4ac (Fig. 7a–d and Supplementary Fig. 5). We found that either SIRT2-treated or untreated p300₍ᵦᵣₚₕz₎ robustly acetylates H3K27 in NCP but produces H3K9ac and H3K4ac to a lesser degree, which is in agreement with our previous findings[25] and with the data shown in Fig. 6j—the reaction was fast and completed in 2 min. The selectivity of p300₍ᵦᵣₚₕz₎ toward the H3K27 site was conserved in the suspension of SIRT2-treated p300₍ᵦᵣₚₕz₎ droplets, however,

again, the rate of NCP acetylation was decreased (Fig. 7b, d, yellow line). These data further substantiate a reduction in the catalytic activity of p300₍ᵦᵣₚₕz₎ upon formation of the condensates.

**p300 condensates preferably localize to chromatin regions marked by H3K27me3.** p300 often binds to enhancers and promoters enriched in active mono- and trimethylated H3K4 (H3K4me1 and H3K4me3, respectively) marks, acetylating histones, and stimulating gene transcription. However, a pool of p300/CBP with a suppressed HAT activity has been shown to associate with poised and silent genomic regions marked by the repressive modification H3K27me3[25,29–31,48,49]. Indeed, ChIP-seq analysis of the H1299 cells expressing Flag-tagged WT p300₍ᵦᵣₚₕzₜ₎ identified 679 p300₍ᵦᵣₚₕzₜ₎ binding sites, and of these,

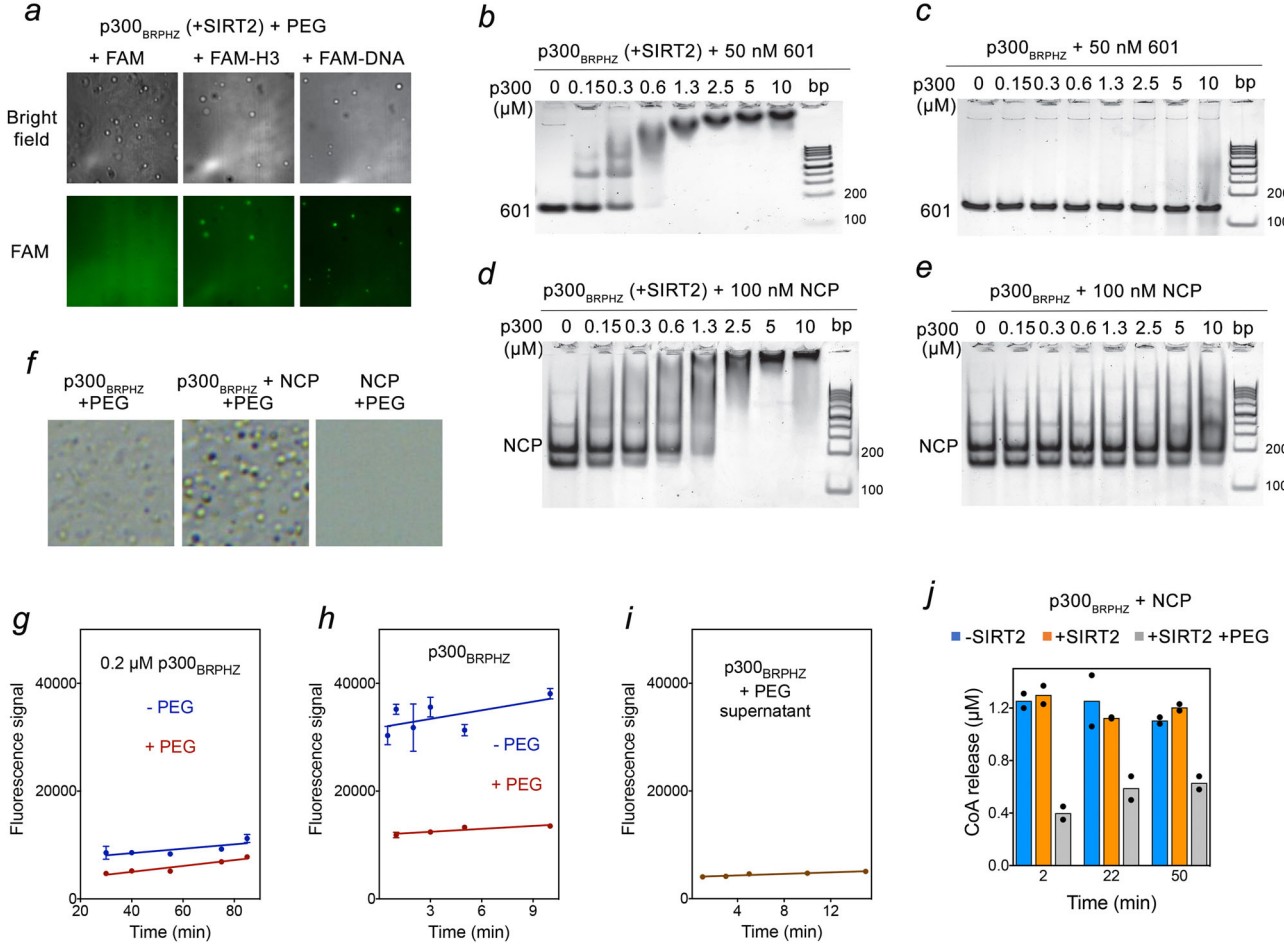

**Fig. 6 Acetylation regulates the association of p300 with DNA and the nucleosome. a** Representative confocal images of phase-separated SIRT2-treated p300$_{BRPHZ}$ condensates with FAM control (left), FAM-labeled H3$_{1-12}$ peptide (middle), and FAM-labeled DNA (right). A 50 μm × 50 μm square region is shown for each sample. **b, c** EMSA of 147-bp 601 DNA incubated with increasing amounts of SIRT2-treated (**b**) and untreated (**c**) p300$_{BRPHZ}$. DNA and protein concentrations are shown above the gel image. **d, e** EMSA of the reconstituted nucleosome incubated with increasing amounts of SIRT2-treated (**d**) and untreated (**e**) p300$_{BRPHZ}$. Nucleosome core particle (NCP) and protein concentrations are shown above the gel image. **f** Phase separation of SIRT2-treated p300$_{BRPHZ}$ alone (left), p300$_{BRPHZ}$ incubated with NCP (middle) and NCP alone (right) in a buffer containing 12% PEG. A 50 μm × 50 μm square region is shown for each sample. Experiments in **a–f** were performed in at least two replicates. **g** HAT activity of diluted (0.2 μM) (**g**) and concentrated (10 μM) (**h**) SIRT2-deacetylated p300$_{BRPHZ}$ ± PEG was measured by a fluorometric assay. **i** HAT activity of SIRT2-deacetylated p300$_{BRPHZ}$ in supernatant measured by a fluorometric assay. Droplets in the phase-separated p300$_{BRPHZ}$ sample were spun down for 10 min, and the supernatant was transferred to another tube. For all experiments in (**g–i**), reactions were quenched by adding 6 M guanidine hydrochloride. The relative reaction rates (slope values) are 0.020 (−PEG) and 0.027 (+PEG) in (**g**), 0.27 (−PEG) and 0.09 (+PEG) in (**h**), and 0.035 in (**i**). Data in **g, h** are presented as mean values ± SD; error bars represent SD from triplicate measurements. **j** HAT activity of untreated p300$_{BRPHZ}$ (blue), SIRT2-treated p300$_{BRPHZ}$ (orange), and SIRT2-treated phase-separated p300$_{BRPHZ}$ droplet suspension (gray) on NCP measured by a fluorometric assay. Data are presented as mean values from duplicate measurements.

117 p300$_{BRPHZT}$ binding sites were also enriched in H3K27ac and H3K18ac—the primary products of acetylation by p300 (Fig. 7e). 76 p300$_{BRPHZT}$ binding sites however were enriched in H3K27me3. The notable absence of acetylation of H3K18 at these sites suggests that the HAT activity of p300$_{BRPHZT}$ is decreased when p300 colocalizes with H3K27me3.

Because the HAT activity of p300 is decreased in the phase-separated condensates, we examined whether the condensates could select for chromatin modifications in HeLa cells using immunofluorescence. As shown in Fig. 8, the YFP-p300$_{FL}$ condensates co-localize to a higher degree with the regions enriched in H3K27me3 with a Pearson's correlation coefficient (Pearson's R) of 0.45 ± 0.09, but co-localization with H3K4me3 was less pronounced (Pearson's R of 0.37 ± 0.09), and even lower level of co-localization was observed with H3K9me3 (Pearson's R of 0.25 ± 0.05). Together, these data suggest that a

pool of p300 condensates has a preference to localize to the chromatin regions containing transcriptionally repressive H3K27me3 modification as compared to the regions containing transcriptionally active modification H3K4me3 or the heterochromatin mark H3K9me3.

It is becoming increasingly clear that the LLPS phenomenon and the assembly of membraneless condensates by biological macromolecules in the nucleus play a crucial role in numerous cellular processes. Formation of condensates allows for efficient separation of the nuclear compartments in a spatiotemporal manner and/or concentration of the macromolecules to regulate, activate or reduce their functions. Although the phase separation mechanisms in the nucleus are currently a subject of intense studies, multivalent weak contacts involving IDRs of macromolecules have been widely acknowledged as a driving force for the formation of biomolecular condensates. In this work, we show

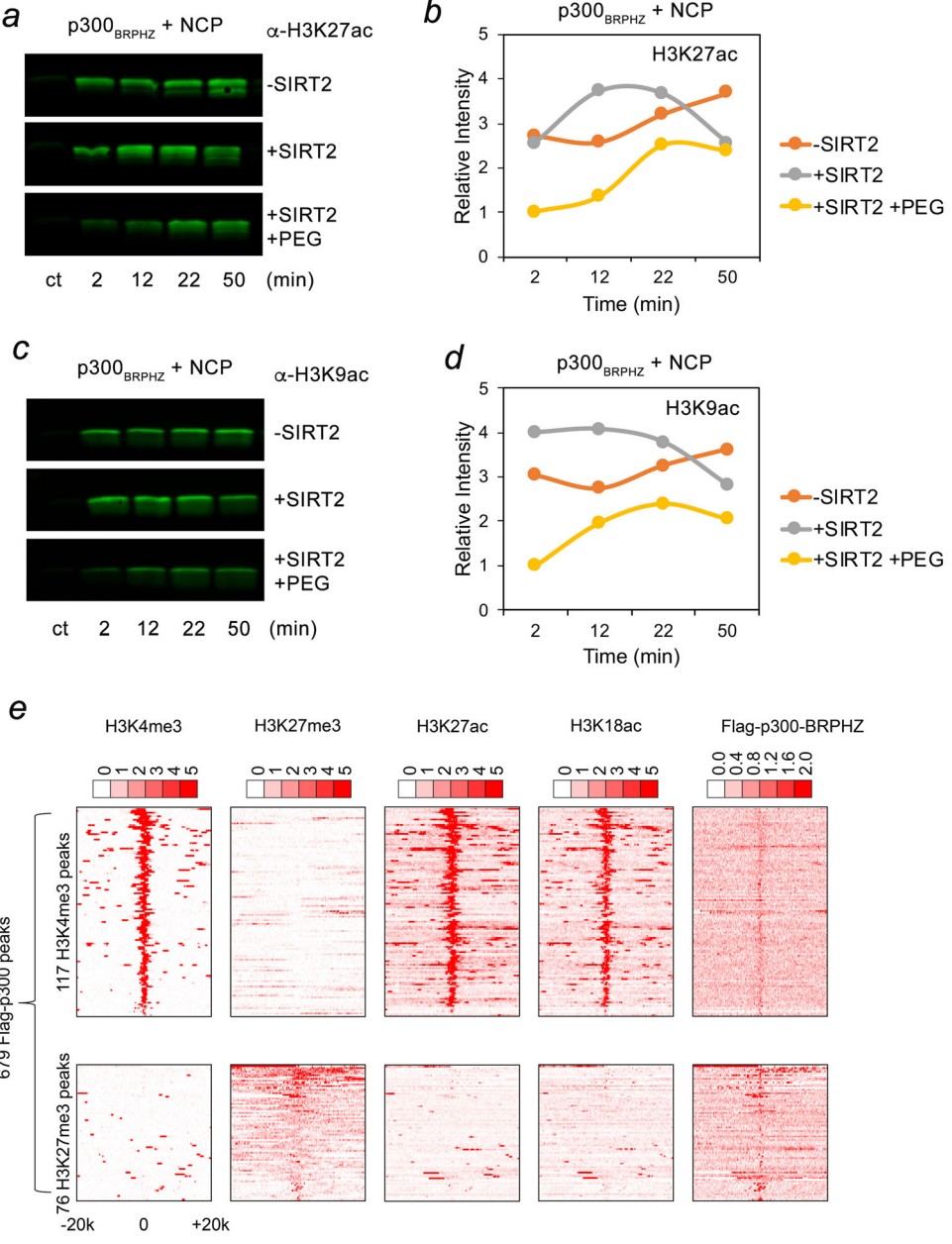

**Fig. 7 A pool of p300 co-localizes with H3K27me3. a–d** H3K27ac and H3K9ac western blot analysis of the reaction mixtures containing p300$_{BRPHZ}$ and an equimolar amount of NCP as a substrate. Reactions were quenched by rapid freezing and the addition of SDS-loading buffer at indicated time points. The intensity of bands is quantified, normalized to the +SIRT2/+PEG band at 2 min, and shown in (**b**) and (**d**). **e** Heatmaps of H3K4me3, H3K27me3, H3K27ac, H3K18ac, and Flag-p300$_{BRPHZT}$, ChIP-seq signals centered on p300-H3K4me3 and p300-H3K27me3 binding sites in a ±20 kb window in H1299 cells stably expressing FLAG-p300$_{BRPHZT}$. Heatmaps are ranked by H3K4me3 (upper panel) or H3K27me3 (lower panel). The color keys represent ChIP-seq densities normalized to total reads.

that the major human acetyltransferase p300 forms liquid condensates in the nucleus and this function depends rather on the structured catalytic core of p300, including the HAT domain and its AIL, and BD. We demonstrate that hyperacetylation of the p300 catalytic core, particularly of AIL, decreases the phase separation ability, and that p300 utilizes two distinct molecular mechanisms to assemble the condensates, which rely on intermolecular 'in trans' HAT–AIL and BD–(Kac outside AIL) interactions. Furthermore, we found that the catalytic HAT activity of p300 is decreased in the phase-separated droplets, which is likely due to steric blocking of the HAT active site.

Our data suggest a model for compartmentalization and concentration of p300 with reduced catalytic activity. p300/CBP is

often associated with transcriptional activation and occupies gene promoters and enhancer elements but has also been reported to localize to the repressive sites, particularly those enriched in H3K27me3[29,30]. These sites are characterized by overall low acetylation of histones, and since H3K27me3 does not preclude binding of p300/CBP to the H3 tail, it was proposed that the HAT activity is blocked at such sites[31,49]. The mechanism of this blockage remains unclear. A few concepts have been put forth to explain the decrease in the p300/CBP catalytic activity, especially on histone lysine residues other than H3K27, as methylation of H3K27 obviously prevents its acetylation. These include regulation through phosphorylation or SUMOylation of p300/CBP[34,50] and rapid degradation of p300. Our finding that a large portion of

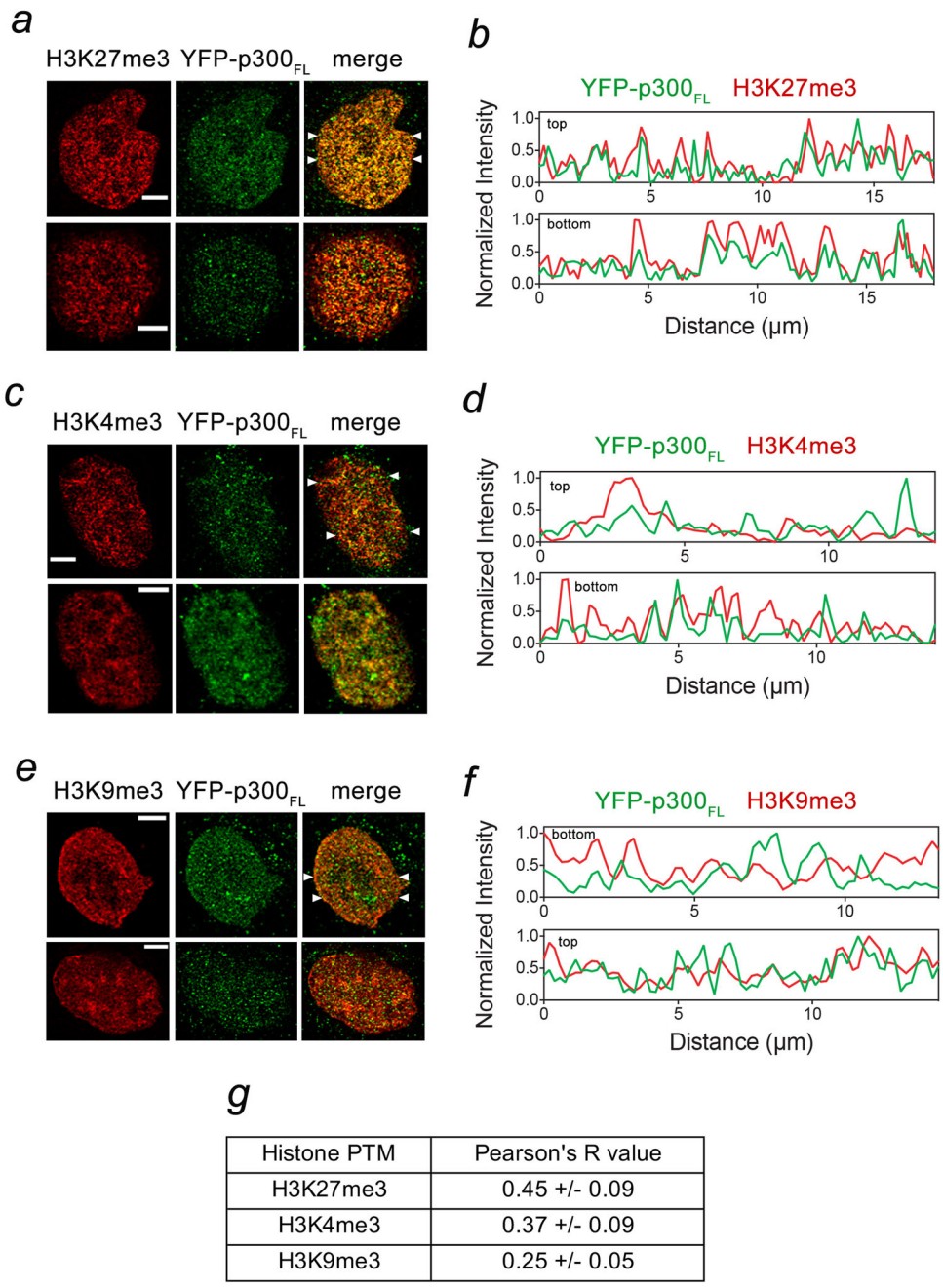

**Fig. 8 p300 condensates show preference for co-localization with H3K27me3. a–f** Nuclear localization of the YFP-p300$_{FL}$ condensates with indicated posttranslational histone modifications (PTMs) in HeLa cells were examined by immunofluorescence. The cells were fixed and probed with anti-H3K27me3 (**a**), anti-H3K4me3 (**c**), and anti-H3K9me3 (**e**) antibodies. Representative images are shown in the left panels: green, YFP-p300$_{FL}$; red, PTMs; yellow, overlap of fluorescence signals of YFP-p300$_{FL}$ and PTMs. The ImageJ software was used to plot fluorescence signal intensities of YFP-p300$_{FL}$ and PTMs in the nuclei along the line (from left to right) indicated by white arrowheads (**b**, **d**, **f**). Scale bar, 5 µm. **g** A table of Pearson's *R* values calculated using the entire cell nuclei (*n* = 3 cells examined for each mark). Data are presented as mean values ± SD. Pearson's *R* values range between +1 (perfect positive correlation) and −1 (inverse correlation).

p300 condensates co-localizes with chromatin regions enriched in H3K27me3 suggests an alternative mechanism. The p300 condensates can act as a storage pool of the protein with reduced HAT activity, allowing p300 and possibly other elements of the transcriptional machinery to be compartmentalized and concentrated at the repressed chromatin sites. The formation of p300 condensates through blocking the catalytic site of the HAT domain provides a mechanism by which the enzymatic activity of p300 can be downregulated. Similar downregulation of the enzymatic activity through phase transition has been reported for the target of rapamycin (TOR) serine/threonine protein kinase in the TORC1 complex[45]. The formation of the TORC1 foci results in steric occlusion of the TOR active site, subsequently leading to the inhibition of the catalytic activity[45]. To better understand the p300-dependent activation of gene transcription, it will be essential in future studies to delineate and visualize by single-molecule imaging[51] the precise contacts between p300 and the repressed chromatin regions.

## Methods

**Cell culture, transfection, and imaging.** WT full-length YFP-p300 plasmids were transfected into HeLa cells in a 3.5-cm-diameter tissue culture dish by Lipofectamine 3000 (Life Technology, L3000-075) using manufacturer instructions. The cells were cultured in DMEM (company) supplemented with 10% fetal bovine serum (company, need double-check) for 36 h. The cell culture medium was replaced with the live-cell imaging medium and maintained at 37 °C using a heater controller. The p300 BD–Ring–PHD–HAT–ZZ_TAZ2 region (BRPHZT, aa 1024–1830) was cloned into a pTripZ-YFP vector. To establish a stable cell line, viral titer harvested from HEK293T cells was used to infect HeLa cells with fusion gene. HEK293T cells were seeded in a 10-cm dish to reach 90–100% confluency the following day. The cells were then transfected using calcium phosphate pre-precipitation (21.0 µg psPAX2, 10.5, 21.0 µg of pTripZ-YFP-p300, 250 mM CaCl₂, and 1×Hank's balanced salt solution) and incubated for 12 h at 37 °C and 5% CO₂. Cells were washed twice with culture medium and then incubated in culture medium for 48 h. Viral titer was harvested from HEK293T cell culture medium and was spun at 1000 × g for 5 min to remove any cell debris. This was added to a single-cell suspension of HeLa cells supplemented with 8 µg/mL Polybrene (SIGMA, H9268) and was mixed. Solution of virus and HeLa cells was plated evenly into a 10-cm dish and incubated for 12 h. Culture medium of infected cells was then replaced, and cells were maintained.

**FRAP experiments.** FRAP imaging was performed using a Zeiss LSM 700 Observer as described previously[16]. Briefly, two images were taken before photobleaching, and 15–30 images were taken with 10 s intervals immediately after photobleaching. The images were analyzed using ImageJ. Fluorescence intensities were normalized to the signal before photobleaching to obtain the fluorescence recovery.

**Protein expression and purification.** The construct of human full-length p300 protein-containing BD–RING–PHD–HAT–ZZ region (BRPHZ, amino acids 1035–1720) was cloned into the pGEX-6P-1 vector and expressed in BL21 (RIL) cells as previously described[25]. Protein production was induced with 0.1 mM IPTG and cultured overnight at 16 °C in Luria broth (LB) medium. The GST-tagged proteins were purified on Pierce glutathione agarose beads (Thermo-Fisher) in 20 mM Tris–HCl (pH 7.5), 500 mM NaCl, and 3 mM DTT. The GST tag was cleaved overnight at 4 °C with PreScission proteases. The SIRT2 (38–356) construct fused with His6x-SUMO was obtained from Addgene (addgene ID:102622). SIRT2 protein was expressed and purified using a standard protocol. In brief, the cells were disrupted by sonication, and the cell lysate was spun down to remove the debris. The supernatant was loaded to a nickel column (Histrap, GE Healthcare) pre-equilibrated with a buffer containing 50 mM Tris–HCl pH 7.5 and 500 mM NaCl. The SIRT2 protein was eluted with a 0–500 mM linear gradient of imidazole. The deacetylation reaction was carried out by incubating purified p300 WT and mutant proteins with SIRT2 overnight in the cold room in a buffer containing 20–50 mM Tris (pH 7.5), 300 mM NaCl, 5 mM MgCl₂, 2 mM NAD, and 2 mM DTT. Deacetylated p300 proteins were further purified by size-exclusion chromatography and concentrated in Millipore concentrators.

**Trypsin digestion of acetylated p300 proteins.** Purified p300_BRPHZ (40–50 µg) with or without SIRT2 treatment was denatured, reduced, and alkylated in 5% (w/v) sodium dodecyl sulfate (SDS), 10 mM tris (2-carboxyethyl) phosphine hydrochloride (TCEP–HCl), 40 mM 2-chloroacetamide, 50 mM Tris pH 8.5 and boiled at 95 °C for 10 min. Samples were prepared for mass spectrometry analyses using the SP3 method[52]. Carboxylate-functionalized speedbeads (GE Life Sciences) were added to protein samples. Acetonitrile was added to 80% (v/v) to precipitate protein and bind it to the beads. The protein-bound beads were washed twice with 80% (v/v) ethanol and twice with 100% acetonitrile. Lys-C/Trypsin mix (Promega) was added for 1:50 protease to protein ratio in 50 mM Tris pH 8.5 and incubated rotating at 37 °C overnight. To clean up tryptic peptides, acetonitrile was added to 95% (v/v) to precipitate and bind peptides to the beads. One wash with 100% acetonitrile was performed and tryptic peptides were eluted twice with 1% (v/v) trifluoroacetic acid (TFA), 3% (v/v) acetonitrile in water. Eluate was dried using a speed-vac rotatory evaporator.

**Liquid chromatography and tandem mass spectrometry (LC–MS/MS) analysis.** For estimations of intact protein masses, untreated and SIRT2-treated WT and mutant p300 protein samples were resolved using a Waters AQCUITY UPLC. Proteins were diluted to final concentration of 0.3 µg/µL using Buffer A (0.1% formic acid in water), of which 3 µg of protein was loaded onto a Waters ACQUITY UPLC Protein BEH C4 Column (300 Å, 1.7 µm, 2.1 mm × 100 mm). Salts were removed with 3% Buffer B (0.1% formic acid in acetonitrile) at 0.2 mL/min for 3 min and proteins were eluted using a linear gradient of Buffer B from 3% to 80% in 4 min at 0.2 mL/min. The UPLC was coupled directly with a Synapt G2 HDMS qTOF mass spectrometer scanning 400–2500 $m/z$. Charge deconvolution was performed using Waters MaxEnt software.

For acetylation sites analysis, the trypsinized peptides were resuspended in 0.1% TFA, 3% acetonitrile in water, of which 1 picomole of the peptides for each sample was directly injected onto a Waters M-class column (1.7 µm, 120 A, rpC18, 75

µm × 250 mm) and gradient eluted from 2% to 40% acetonitrile over 40 min at 0.3 µL/min using a Thermo Ultimate 3000 UPLC (Thermo Scientific). Peptides were detected with a Thermo Q-Exactive HF-X mass spectrometer (Thermo Scientific) scanning MS1 spectra at 120,000 resolution from 380 to 1580 $m/z$ with a 45 ms fill time and 3E6 AGC target. The top 12 most intense peaks were isolated with a 1.4 $m/z$ window with a 100 ms fill time and 1E6 AGC target and 27% HCD collision energy for MS2 spectra collected at 15,000 resolution. Dynamic exclusion was enabled for 5 s. MS data raw files were searched against the single Uniprot sequence for EP300 (Uniprot accession number Q09472) using Maxquant 1.6.14.0 with cysteine carbamidomethylation as a fixed modification, while methionine oxidation and protein N-terminal and lysine side chain acetylation were set as variable modifications. The mass tolerances for the database search were 4.5 ppm for the precursors and 20 ppm for the MS2 fragment ions, the minimum peptide length was 7 residues with no additional applied score cutoffs. Peptide and protein level FDR was set at 0.01. An intensity cut-off of 50 million counts was applied to the untreated sample prior to plotting the change in intensity in Fig. 2j.

**In vitro condensate formation.** All in vitro condensate formation assays were performed in a buffer containing 15 mM Tris–HCl (pH 7.5), 150 mM NaCl and 2 mM DTT unless otherwise stated. All samples were prepared on ice and incubated for ~5 min before imaging on siliconized glass cover slides (Hampton). For WT and mutant p300_BRPHZ and p300_HZ alone, 13 µM protein samples were parallelly prepared with or without 12% (w/v) PEG 3350. For the condensate formation assay with reconstituted nucleosomes, 6.4 µM SIRT2-treated p300_BRPHZ and an equal amount of NCP were incubated in a buffer containing 10 mM Tris 7.5, 60 mM NaCl, 2 mM DTT, and 12% (w/v) PEG350. Microscopy of the droplets was done using an M150C-I microscope (AmScope) equipped with a ×10 objective and an MD35 digital camera (AmScope). A microscope camera calibration slide (OMAX, 0.01 mm) was used to determine the scale. A 50 µm × 50 µm square area was selected as a representative image for each sample. The number of condensates was also counted in a 50 µm × 50 µm square area. Five non-overlapping square regions were counted for each sample and plotted. Experiments were repeated in at least three batches of purified and SIRT2-treated p300 proteins.

To prepare p300 condensates that concentrate DNA or H3, 13 µM SIRT2-treated p300 was mixed with either 6 µM FAM, 6 µM FAM-labeled 37 bp dsDNA, or 6 µM FAM-labeled histone H3 tail (aa 1–12). Confocal images were acquired on a Zeiss Observer.Z1 inverted microscope using a ×40 oil objective and digitally captured. For the excitation of FAM, a 488 nm laser was used. Images were processed and presented using ImageJ and Photoshop.

**Nucleosome reconstitution.** The poly-cistronic vector of Xenopus laevis Histone octamers was obtained from the Jean-Francois Couture lab. A His-tag and a TEV cleavage site were introduced before histone H3 by mutagenesis. Expression and purification of histone octamers were performed as reported previously[53]. The His-tag before histone H3 was cleaved to expose the Ala1 residue using TEV protease. 601 DNA was prepared as previously described[54]. NCPs were reconstituted by combining octamer with 1.1× excess DNA and performing slow salt dialysis. Reconstituted nucleosomes were further purified by size-exclusion chromatography and concentrated in Millipore concentrators.

**Acetyltransferase assays.** Purified untreated and SIRT2-treated p300_BRPHZ proteins were buffer exchanged into reaction buffer containing 10 mM Tris–HCl (pH 7.5) and 60 mM NaCl. For auto-acetylation experiments in Fig. 3a–c, 10 µM p300 protein was incubated in HAT reaction buffer (10 mM Tris pH 7.5 and 60 mM NaCl) in the absence or presence of 12% (w/v) PEG 3350. For histone acetyltransferase assays in Fig. 6j, 6.4 µM p300 protein, and 6.4 µM reconstituted NCP were incubated in HAT reaction buffer (10 mM Tris pH 7.5 and 60 mM NaCl). Reactions were started by adding 0.5 mM acetyl-CoA at room temperature and quenched by flash-freeze at indicated time points. For each time point, 8 µl reaction mixture was diluted to 200 µl assay buffer and immediately heated to 95 °C for 5 min to inactivate p300. The samples were then applied to an ultrafiltration device (10k cut-off) to collect flow-through for CoA quantification by the fluorometric assay kit (Abcam, 138889). The kit utilizes a fluorogenic green indicator that became strongly fluorescent upon reacting with the –SH group in CoA. The fluorescence signals in samples collected from the reactions were measured according to the product manual using a 96-well microplate reader. For measuring the HAT activity of p300 (Fig. 6g–i) in the diluted solution, droplet mixture, and supernatant, reactions were quenched by mixing with a denaturing buffer containing 20 mM Tris–HCl (pH 7.5), 150 mM NaCl and 6 M guanidine–HCl at indicated points. All experiments were performed in triplicates.

To detect histone H3 acetylation at specific lysine sites, reactions were quenched by flash-freezing in liquid nitrogen and then analyzed by SDS–PAGE and western blot analysis. Western blot results were quantified by LI-COR Odyssey System using the following antibodies: anti-H3K4ac (ab176799, 1:1000), anti-H3K9ac (ab4441, 1:1000), and anti-H3K27ac (ab177178, 1:1000) from Abcam, and anti-H3K18ac (39755, 1:1000) from Active Motif.

**SAXS analysis.** X-ray scattering data were collected at the Bio-SAXS beamline (BM29) of the European Synchrotron Radiation Facility. Data were collected with a

photon-counting Pilatus 1 M detector at a sample-detector distance of 2.86 m, a wavelength of $\lambda = 0.991$ Å, and an exposure time of 1 second/frame. A momentum transfer range of 0.008–0.47 Å$^{-1}$ was covered ($q = 4\pi \sin \theta/\lambda$, where $\theta$ is the scattering angle and $\lambda$ the X-ray wavelength). A time-resolved SAXS experiment was performed with the p300 (aa 324–2094) and ΔAIL p300 constructs, produced as described[55], during auto-acetylation. 150 μL of the reaction mixtures in buffer (20 mM HEPES pH7.0, 500 mM NaCl, 5 μM ZnCl$_2$, 0.5 mM TCEP) were prepared and pipetted in the thermostated sample holder tube already adjusted to 30 °C. 2 mM acetyl-CoA was added to each reaction tube and in the background, buffer to start the reaction. Four scattering curves were recorded on each sample at 0, 30, 60, and 90 min. $R_g$ and $I(0)$ values were obtained from the Guinier approximation $R_g < 1.3$ using Primus[56]. Distance distribution functions $p(r)$ were computed from the entire scattering curve using GNOM[56].

**EMSA with DNA and nucleosomes**. EMSA experiments were performed essentially as described[57]. Briefly, increasing amounts of p300 were incubated with 147 bp 601 DNA (50 nM) in a DNA binding buffer containing 20 mM Tris–HCl (pH 7.5), 50 mM NaCl, and 2 mM DTT for 5 min. The reaction mixtures were loaded on 5% polyacrylamide gels, and electrophoresis was performed in $0.2 \times$ TBE buffer at 100 V for 1.5 h on ice. For EMSA experiments with reconstituted nucleosomes (100 nM), the NaCl concentration in binding buffer was increased to 75 mM, and electrophoresis was performed in $0.2 \times$ TB buffer. Gels were stained with SYBR Gold (Invitrogen).

**ChIP-seq analysis**. Flag-p300$_{BRPHZT}$, H3K18ac, and H3K27ac ChIP-seq reads were obtained from GSE109591; H3K4me3 and H3K27me3 ChIP-seq reads were from GSE81322. ChIP-seq raw reads were mapped to the hg38 genome by hisat2 (v2.1.0) with no-spliced-alignment, -k 1. H3K4me3 and H3K27me3 peaks were called by macs2 callpeak (v2.1.2) with broad parameter. Flag-p300$_{BRPHZT}$ peaks were converted to hg38 by UCSC liftOver. Heatmaps were generated by danpos (v2.2.2) with 200 bp bin size and visualized by TreeView (v1.1.6).

**Immunofluorescence**. HeLa cells were cultured in a six-well plate with 22-mm coverslips. YFP-p300 plasmids were transfected into HeLa cells by Lipofectamine 3000 (Life Technology, L3000-075) according to manufacturer instructions. One day after transfection, cells were fixed by 1% paraformaldehyde and then permeabilized with 0.2% Triton X-100. After blocking with 3% goat serum and 3% BSA, cells were incubated, respectively, with pairs of primary antibodies: anti-GFP (Life Technologies; A-11120; 1:400 dilution) and anti-H3K27me3 (Millipore; 07-449; 1:200 dilution), anti-GFP (Life Technologies; A-11120; 1:400 dilution) and anti-H3K9me3 (Upstate; 07-442; 1:200 dilution), and anti-GFP (Life Technologies; A-11120; 1:400 dilution) and anti-H3K4me3 (Novus; NB21-1023B; 1:200 dilution), for two hours. After washing, cells were incubated with a pair of secondary antibodies: Alexa Fluor 488-labeled goat anti-mouse and Alexa Fluor 568-labeled goat anti-rabbit, for two hours, and then were mounted by using ProLong Antifade reagents (Life Technologies; P7481). Line plots were made using the plot profile feature of ImageJ. Pearson coefficient correlation for each pair of immunofluorescence images was calculated using the Coloc 2 plugin in ImageJ after the selection of the entire cell nuclei as the regions of specific interest.

## Data availability
All relevant data supporting the key findings of this study are provided in the Supplementary Information and Source Data files or from the corresponding author upon reasonable request. The mass spec data generated in this study have been deposited to the PRIDE database under the accession number PXD026898. The publicly available ChIP-seq data analyzed in this study are available from Gene Expression Omnibus under accession codes GSE109591 and GSE81322. Source data are provided with this paper.

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

## Acknowledgements

We thank Dr. Berk for providing cDNA of YFP-p300. The authors thank Cole Michel at the University of Colorado School of Pharmacy Mass Spectrometry Facility for help with collecting and analyzing samples. This work was supported by grants from NIH GM125195, GM135671, HL151334, CA252707, and AG067664 to T.G.K., R01GM135286 to X.R. and CA204020 to X.S. Y.Z. is supported by NIH K99CA241301.

## Author contributions

Y.Z., K.B., Y.Y., Z.I., M.Z., H.X., S.I., T.L., C.C.E., and J.L. performed experiments and together with D.P., X.S., X.R., and T.G.K. analyzed the data. Y.Z. and T.G.K. wrote the manuscript with input from all authors.

## Competing interests

The authors declare no competing interests.
