## [Peer Review File · Nature Communications]

REVIEWER COMMENTS

Reviewer #1 (Remarks to the Author):

The assembly of liquid condensates by biological macromolecules (liquid-liquid phase separation) plays a crucial role in numerous cellular processes. In this work, authors revealed that p300, a major human acetyltransferase, can form liquid condensates in the nucleus. Authors further showed that HAT domain and its autoinhibitory loop (AIL), and bromodomain (BD) is important for formation of liquid condensates. Acetylation level of AIL is negatively correlated with P300 phase separation ability. Authors suggested that p300 utilizes two distinctive molecular mechanisms to assemble the condensates, which rely on intermolecular 'in trans' HAT-AIL and BD-(Kac outside AIL) interactions. They also found that the catalytic HAT activity of p300 is decreased in the phase separated droplets and enriched with H3K27Me3. Taken together, they work suggest a model in which p300 condensates can act as a storage pool of the protein with reduced HAT activity, allowing p300 to be compartmentalized and concentrated at poised or repressed chromatin regions. The manuscript is well-written and the presentation is clear. The conclusions are supported by solid data. It would deserve being considered for publication, should the authors appropriately address the following points:

Questions:

1. Abstract: Author made a strong claim saying that “p300 is regulated by autoacetylation and relies on its catalytic core, representing a rare example of biomolecular phase separation occurred through structured regions - the catalytic HAT domain, the autoinhibition loop, and bromodomain”. However, the key determinant the long autoinhibition loop is largely intrinsic disordered, therefore, P300 is not an exception from what we have known in the literature. Please tune down the statement.
2. Page 5, authors showed that both Full-length (FL) p300 and truncated p300 can form puncta in cells. Puncta formation depends on concentration of proteins. Have authors compared core p300 vs FL- p300 using the same expression vectors or compare in vitro droplet formation assays? Since FL- p300 and truncated-p300 are expressed in two different vectors, it is not clear the exact concentration of these proteins in the cells. This information would be informative to understand relative contributions of core domain vs flanking regions.
3. Page 5. the authors postulated the core domain might be involved in the phase separation process. In most cases, the intrinsic disordered regions are involved in the phase separation. Have authors examined the roles of flanking regions in phase separation as well?
4. Are ZZ domain, PHD domain or Ring domain necessary for condensate formation of p300? Authors seems not discuss the roles of these domains in modulating condensate formation in the paper.
5. Page 10, authors showed that p300BRPHZ condensates can sequester nucleosomal components like histone H3 tail and DNA from surrounding environment. Are they through non-specific interactions? Authors suggested that H3 tail may be sequestered maybe through ZZ domain. This prediction can be test in a straightforward manner with p300 truncate without ZZ domain. In

addition, have authors tested other histone tails (as controls) to see whether the observed sequestration of histone tails is specific to H3 or not?

6. Page 11, authors showed that addition of NCP promotes formation of p300 droplets. Can 601 DNA alone able to facilitate formation of droplets of p300 ?

Reviewer #2 (Remarks to the Author):

Paper Summary:

In this paper, evidence is presented for the condensation of p300, which is an acetyltransferase. Further in vitro experiments reveal a mechanistic basis for this condensation, which can occur via two possible mechanisms, and is attenuated by acetylation of the AIL subunit. Condensation is suggested to have functional implications in three separate ways: 1. Condensates sequester nucleosome-associated biomolecules in vitro; 2. Catalytic activity of p300 is decreased upon formation of condensates in vitro; and, 3. p300 condensates localize to chromatin marked with H3K27me3 in vivo.

Review Summary:

The authors present a manuscript exploring an interesting example of condensation in a (mostly) protein system, with potentially functional consequences. These findings are likely to be of interest to a broad community, given the ubiquity of p300 interactions, the importance of acetylation in regulation, and the continued interest in condensation/LLPS in biology. The novelty of the findings is enhanced since the interactions leading to condensation are between structured elements of the proteins, in contrast to other common examples of biomolecule condensation. The manuscript itself is well-organized and clearly argued.

For these reasons, I think this manuscript is acceptable for publication in Nature Communications. Some fairly minimal changes are encouraged; these changes are detailed below.

Specific Comments:

1. In Fig. 2i, the placement of all four conditions equally spaced along the same axis feels misleading, for two reasons. First, the intervals between approximate Kac levels are not quantitatively even. Second, my understanding is that the 26 and 16 Kac conditions represent qualitatively distinct conditions from the 12 and 4 Kac conditions. In the latter, SIRT2 treatment preferentially de-acetylates K residues in the AIL. Since (one of) the purported mechanism of condensation relies on the interaction of the AIL with HAT, it does not seem correct to compare conditions where residues in the AIL have been specifically de-acetylated with conditions which might have more dispersed de-acetylation. Can the authors re-plot these data in ways that avoid conflating these two conditions?

2. In Fig. 3e, the authors draw a diagram which suggests that condensation does not occur because of an interaction that occurs between BD and KacAIL in cis. Even if cis BD-KacAIL interactions compete with trans BD-KacAIL interactions, it would still suggest that there is on average a slight attraction between multiple autoacetylated WT p300. This could lead to condensation at higher concentrations even under autoacetylated conditions. Can the authors either provide an explanation as to why trans BD-KacAIL interactions are not present, or perform additional experiments investigating the presence of condensation at higher concentrations?

3. In Figs. 6g-i, the primary evidence is the change in slope upon the addition of PEG in the fluorescence measurements. This is difficult to assess visually – can the authors provide numbers to give more confidence to this somewhat subtle visual effect? Additionally, I personally found the current explanation of the link between fluorescence slope and catalytic activity in the text too terse. I think the manuscript would be improved with a small amount of additional text which summarizes the relevant experimental principles.

4. In Fig. 8, the authors report on r-values of colocalization of histone marks and YFP-p300 in sections of a nucleus. This is done for two horizontal slices per histone mark. The authors use the results of this analysis to assert that p300 preferentially associates with the repressive mark H3K27me3, but I think additional analysis is necessary, particularly in comparing H3K27me3 and H3K4me3. These two marks showed a marked variation in the two r-values reported, with values that are not that distinct. It is also not visually clear to me whether it is fair to say that p300 is “preferentially” associated with H3K27me3 as opposed to H3K4me3. The authors should compute additional r-values to provide more confidence in their results. When doing so, they should additionally avoid slices such as the top slice in 8a. On that slice, it appears as if a large nuclear body is excluding both p300 and H3K27me3 – this can inflate correlation for reasons that likely have nothing to do with the claims at hand.

5. The authors should additionally be careful to not overstate the conclusions they reach from their in vitro evidence that condensation slows down p300’s catalytic activity. For example, in line 323, the authors state that “Because the HAT activity of p300 is decreased in the phase separated condensates, we examined whether the condensates select for a specific chromatin modification in HeLa cells using immunofluorescence.” While in vitro this is true, the authors do not test if other factors in the nuclear environment might counterbalance the reduction in activity associated with condensation seen in vitro. Lacking a clear mechanistic picture of where the condensation-associated activity reduction comes from, the authors should be more careful in stating the functional implications of their observations.

Reviewer #3 (Remarks to the Author):

“Nuclear condensates of p300 formed through the structured catalytic core can act as a storage pool of p300 with reduced HAT activity” by Zhang et. al.

This paper attempts to show that p300 phase separates and that this is a form of regulation. This regulation is itself regulated by autoacetylation mostly in trans. This will be an important finding assuming the controls and other points can be addressed.

Major concerns:

Are the same autoacetylation sites observed on full-length p300?

In the autoacetylation assay result in equivalent moles of CoA to new acetylation sites on p300(BRPHZ)?

If there is competition between the bromodomain and acetyl-AIL than shouldn't a bromodomain inhibitor also change condensation? Or for that matter a acetyl-histone?

If it is capable of autoacetylation why wasn't it already acetylated by the time the experiment was performed? I assume this protein was expressed in cells which have acetyl-CoA? Was it deacetylated and mixed with acetyl-CoA over the course of the experiment? "ΔAIL p300340-2094, the Rg value increased over time (to ~ 35 nm) after autoacetylation for 1.5 hr"

There should be more detail on which MS systems and approaches were used when.

Missing controls:

wt p300 needs to be knocked out in the YFP experiments. You need to show that it is effective as the only source of p300. It is also possible that the phase transition is driven by increased p300 and/or auto-trans acetylation.

If autoacetylation is driving the observed behavior they should all be the same if you add excess acetyl-CoA if not you have a folding issue with your construct.

Given the multiple issues with antibodies it is necessary to at least divide the signal from the PTM AB by a H3 antibody. Or better show it by MS/MS.

Reviewer #4 (Remarks to the Author):

Phase-separation by nuclear proteins has been suggested to regulate a variety of chromatin-based processes. Some commonly proposed roles for such phase-separation are concentration of reagents to increase reaction rates, and sequestration of genomic regions to reduce transcription. Yet, to date studies that directly test the effects of phase-separation on catalysis by chromatin regulators are quite limited. Also limited are studies that dissect specific protein-protein interactions that drive the multi-valency needed for phase-separation. Both types of studies are essential to understand how phase-separation can regulate specific chromatin-based pathways. In this work the authors aim to bridge these two gaps in the context of a major co-activator and histone acetyl transferase (HAT), p300. P300 has several domains flanking the active site in addition to an auto-inhibitory loop (AIL) within the HAT domain. The authors first show data indicating that p300 forms phase-separated droplets in vitro under physiological buffer conditions with PEG as a crowding agent. They then (i) carry out a careful structure-function analysis of the protein-protein interfaces driving p300 phase-separation, (ii) test the role of auto-acetylation and (iii) test the impact of phase-separation on HAT activity. Based on their results they conclude that (i) at least two types of interaction interfaces promote phase-separation, the AIL-active site interaction and an interaction between a flanking

Bromodomain (BD) and auto-acetylated HAT domain residues; (ii) acetylation of the AIL disrupts phase-separation; (iii) the AIL binds DNA and the active site in a mutually exclusive manner; (iv) catalytic activity within phases formed from lowly acetylated p300 is reduced compared to activity in solution and; (v) phase-separation by p300 allows storage of catalytically damped p300 at regions repressed via H3K27me-based mechanisms.

Overall, the experiments are well designed and carefully carried out to allow dissection of a complex system of competing interaction interfaces. The work provides new insights into how co-activators can use distinct interaction interfaces to tune phase-separation in response to post-translational modifications like acetylation. The finding that phase-separation inhibits HAT activity is quite significant as it goes against the common hypothesis that phase-separation increases activity by increasing reagent concentrations. Importantly the authors' new proposed model implies that auto-acetylation of the AIL couples the dissolution of the phase to activation of the enzyme. This mechanism opens an exciting new viewpoint to ask whether other auto-inhibitory domains that are found in multiple chromatin regulators analogously couple regulation of phase-separation to regulation of activity. Finally, the work provides a much-needed example of how systematic biochemical studies can be correlated with cellular phenotypes to provide some clarity on the role of phase-separation in chromatin regulation. For all of the above reasons this work will be of broad interest to scientists from several fields (for example, chromatin, phase-separation and transcription communities).

I support the publication of this work in Nature Communications after the authors address the following few additional experimental suggestions and clarifications, which are expected to strengthen the significance of the work. I anticipate these experiments are likely feasible using reagents already described in the manuscript.

Comments

1. In terms of the model (Fig. 3e), it is not clear why the authors propose that in the hyperacetylated state, the AIL and BD form only "in-cis" and not "in-trans" interactions. Is this based on known structural constraints? Or are they making a local concentration-based argument?
2. The figures with droplets seem not very sharp. For some of the key experiments (eg. Figs. 1f, 4g, 6f), figures showing the droplets more clearly will strengthen the key arguments. Increasing the time before imaging so that the droplets are settled and more of them are in one plane could help.
3. The measurements shown of condensate per 50 um x 50 um area appear be the number of droplets and not the size or volume of droplets. The critical concentration, i.e., the lowest concentration at which phase-separation occurs is typically used in the field to quantify the efficacy of phase-separation. Droplet number (as opposed to cumulative droplet sizes), is more complicated to interpret in terms of phase-separation efficacy as droplet number will be affected by how fast droplets fuse, which in turn will vary based on viscosity effects. Therefore, for the same key experiments mentioned above in #2 a measurement of the saturation/critical concentrations will enable better estimation of the effects of the p300 mutants and of nucleosomes.
4. For Fig. 3c, a control with Co-A + p300-BRPHZ+ PEG is needed to rule out that the Co-A part of

Acetyl -CoA is acting as a hydrotrope (PMID: 28522535).

5. For the nucleosome acetylation assay in Fig. 6j, an additional control is needed to test that PEG does not affect reaction on nucleosomes. This would be analogous to the control in Fig. 6g, where the effect of PEG on auto-acetylation is shown to be negligible. Also, it will be helpful if the authors can speculate why SIRT2 treatment does not affect activity on nucleosomes. Based on the model, it could be expected that the AIL would be better at inhibiting activity in the unacetylated state. Is this because nucleosomes being used are saturating concentrations and that their binding displaces the AIL loop?

6. For the nucleosome acetylation data shown in Fig. 6j, it's possible that the inhibitory effect (grey color) is larger than the ~ 3-fold that can be estimated from the 2 min time-point. This is because the reaction in orange is essentially over by the 2 min time point. If the authors take earlier time points than 2 minutes and plot a rate, they may find that the inhibitory effect is larger than 3-fold, and this would increase the biological significance.

7. Some clarification is needed to understand why the authors conclude that p300 is specific for H3K27 over H3K9 based on the figures in 7a-d. From these figures it is not easily apparent how the extent of H3K27 vs H3K9 acetylation can be directly compared given that different antibodies are used for each mark.

Editorial requests for mass spectrometry analysis:

* For all intact protein mass spectra, please provide the non-deconvoluted spectra in the source data file.

* For all species identified by intact protein LC-MS, please report the theoretical mass, experimental mass and mass error. This can be done in a separate Supplementary Table or by extending the tables that are already in the figures.

* In the Methods section, please explain how the acetylation level quantification in Fig 2j was done. Since you are comparing two independent samples, please also clarify how you controlled for systematic errors (e.g. differences in sample input) and which peptides were compared (only peptides with identical sequence or also peptides with different sequence that point to the same acetylation site? If the latter - how did you account for potential differences in ionization efficiency between these peptides?).

* When describing the acetylation site analysis in the Methods, please also state the mass resolution of the MS1 and MS2 scans and provide more details on the database search parameters and acceptance criteria used for peptide identification (name and version of protein sequence database, mass tolerance for precursor and fragment ions, minimum peptide length, any applied score cutoffs, peptide- and protein-level FDR)

We thank the Editor and Reviewers for the insightful and very constructive comments, which were helpful in revising and strengthening this manuscript.

Reviewer 1, Comment 1: *Abstract: Author made a strong claim saying that “p300 is regulated by autoacetylation and relies on its catalytic core, representing a rare example of ... However, the key determinant the long autoinhibition loop is largely intrinsic disordered, therefore, P300 is not an exception from what we have known in the literature. Please tune down the statement.*

Author's response: as suggested, this statement (representing a rare example of...) has been removed.

Reviewer 1, Comment 2: *Page 5, authors showed that both Full-length (FL) p300 and truncated p300 can form puncta in cells. Puncta formation depends on concentration of proteins. Have authors compared core p300 vs FL- p300 using the same expression vectors or compare in vitro droplet formation assays?*

Author's response: because the p300 core is expressed in the pTripZ vector, whereas FL p300 requires a pcDNA6 vector for expression, we could not compare concentration of proteins, and FL p300 (300 kDa) is too large to be expressed in *E. coli* to carry out *in vitro* assays.

Reviewer 1, Comment 3: *Page 5. the authors postulated the core domain might be involved in the phase separation process. In most cases, the intrinsic disordered regions are involved in the phase separation. Have authors examined the roles of flanking regions in phase separation as well?*

Author's response: we haven't examined other regions of p300 because it is very difficult to obtain/express large p300 constructs. We found that the size of the p300 core itself, which is ~80 kDa, is somewhat at the limit of expression in *E. coli*.

Reviewer 1, Comment 4: *Are ZZ domain, PHD domain or Ring domain necessary for condensate formation of p300? Authors seems not discuss the roles of these domains in modulating condensate formation in the paper.*

Author's response: we found that the PHD and RING domains are not essential. As shown in Fig. 4i, HAT-ZZ without the RING and PHD domains can phase separate, however the presence of AIL is critical. The expression level of the HAT domain itself is very low, therefore we used the HAT-ZZ construct that is expressed well.

Reviewer 1, Comment 5: *Page 10, authors showed that p300BRPHZ condensates can sequester nucleosomal components like histone H3 tail and DNA from surrounding environment. Are they through non-specific interactions? Authors suggested that H3 tail may be sequestered maybe through ZZ domain. This prediction can be test in a straightforward manner with p300 truncate without ZZ domain. In addition, have authors tested other histone tails (as controls) to see whether the observed sequestration of histone tails is specific to H3 or not?*

Author's response: we have previously thoroughly characterized binding of the ZZ domain of p300 to H3 tail (modified and unmodified) and nucleosomes (RING and PHD do not bind to the H3 tail). We determined the structure of the p300 ZZ-H3 complex and showed that the ZZ domain selects for unmodified H3 tail; we refer to this study (ref. 25).

Reviewer 1, Comment 6: *Page 11, authors showed that addition of NCP promotes formation of p300 droplets. Can 601 DNA alone able to facilitate formation of droplets of p300?*

Author's response: DNA alone (we have tested 37bp DNA) does not stimulate formation of more and larger droplets as NCP does.

Reviewer 2, Comment 1: ... In Fig. 2i, the placement of all four conditions equally spaced along the same axis feels misleading, for two reasons. First, the intervals between approximate Kac levels are not quantitatively even. Second, my understanding is that the 26 and 16 Kac conditions represent qualitatively distinct conditions from the 12 and 4 Kac conditions. In the latter, SIRT2 treatment preferentially de-acetylates K residues in the AIL. Since (one of) the purported mechanism of condensation relies on the interaction of the AIL with HAT, it does not seem correct to compare conditions where residues in the AIL have been specifically de-acetylated with conditions which might have more dispersed de-acetylation. Can the authors re-plot these data in ways that avoid conflating these two conditions?

Author's response: as suggested, we have separated the two sets.

Reviewer 2, Comment 2: In Fig. 3e, the authors draw a diagram which suggests that condensation does not occur because of an interaction that occurs between BD and KacAIL in cis. Even if cis BD-KacAIL interactions compete with trans BD-KacAIL interactions, it would still suggest that there is on average a slight attraction between multiple autoacetylated WT p300. This could lead to condensation at higher concentrations even under autoacetylated conditions. Can the authors either provide an explanation as to why trans BD-KacAIL interactions are not present, or perform additional experiments investigating the presence of condensation at higher concentrations?

Author's response: we have shown by SAXS experiment (Fig. 5) that hyperacetylated p300 remains monomeric, implying that the interaction BD-KacAIL *in trans* is less favorable.

Reviewer 2, Comment 3: In Figs. 6g-i, the primary evidence is the change in slope upon the addition of PEG in the fluorescence measurements. This is difficult to assess visually – can the authors provide numbers to give more confidence to this somewhat subtle visual effect? Additionally, I personally found the current explanation of the link between fluorescence slope and catalytic activity in the text too terse. I think the manuscript would be improved with a small amount of additional text which summarizes the relevant experimental principles.

Author's response: as suggested, we have included numbers in Fig. 6g-i legend.

Reviewer 2, Comment 4: In Fig. 8, the authors report on *r*-values of colocalization of histone marks and YFP-p300 in sections of a nucleus. This is done for two horizontal slices per histone mark. The authors use the results of this analysis to assert that p300 preferentially associates with the repressive mark H3K27me3, but I think additional analysis is necessary, particularly in comparing H3K27me3 and H3K4me3. These two marks showed a marked variation in the two *r*-values reported, with values that are not that distinct. It is also not visually clear to me whether it is fair to say that p300 is “preferentially” associated with H3K27me3 as opposed to H3K4me3. The authors should compute additional *r*-values to provide more confidence in their results. When doing so, they should additionally avoid slices such as the top slice in 8a. On that slice, it appears as if a large nuclear body is excluding both p300 and H3K27me3 – this can inflate correlation for reasons that likely have nothing to do with the claims at hand.

Author's response: we have clarified that Pearson's R values were calculated using the entire cell nuclei (not specific lines/slices) in Fig. 8g legend.

Reviewer 2, Comment 5: ... The authors should additionally be careful to not overstate the conclusions they reach from their *in vitro* evidence that condensation slows down p300's catalytic activity. For example, in line 323, the authors state that "Because the HAT activity of p300 is decreased in the phase separated condensates, we examined whether the condensates select for a specific chromatin modification in HeLa cells using immunofluorescence." While *in vitro* this is true, the authors do not test if other factors in the nuclear environment might counterbalance the reduction in activity associated with condensation seen *in vitro*. Lacking a clear mechanistic picture of where the condensation-associated activity reduction comes from, the authors should be more careful in stating the functional implications of their observations.

Author's response: we have toned down this correlation, referring to a pool of condensates, and revised the sentence to '...whether the condensates could select for chromatin modifications...'

Reviewer 3, Comment 1: Are the same autoacetylation sites observed on full-length p300?

Author's response: yes, it was reported by Black et al., 2008 (we cite this ref. #42).

Reviewer 3, Comment 2: If there is competition between the bromodomain and acetyl-AIL than shouldn't a bromodomain inhibitor also change condensation? Or for that matter a acetyl-histone?

Author's response: although there is no competition between bromodomain and acetylated AIL, inhibition of bromodomain should lead to the same result as the mutation of the bromodomain shown in Figs. 4a-c.

Reviewer 3, Comment 3: If it is capable of autoacetylation why was't it already acetylated by the time the experiment was performed? I assume this protein was expressed in cells which have acetyl-CoA? Was it deacetylated and mixed with acetyl-CoA over the course of the experiment? "ΔAIL p300340-2094, the Rg value increased over time (to ~ 35 nm) after autoacetylation for 1.5 hr"

Author's response: The p300 construct (340-2094), produced in insect cells and used in SAXS experiments, is also autoacetylated upon expression but to a lesser degree compared to constructs expressed in *E. coli*, as shown in our previous studies, refs. 28 and 55.

Reviewer 3, Comment 4: There should be more detail on which MS systems and approaches were used when.

Author's response: we have added more detail on MS experiments and analysis in the source data file and methods section.

Reviewer 3, Comment 5: *wt* p300 needs to be knocked out in the YFP experiments. You need to show that it is effective as the only source of p300. It is also possible that the phase transition is driven by increased p300 and/or auto-trans acetylation. – since endogenous and expressed p300/CBP have been shown to form similar nuclear bodies/droplets (compared in refs. 34-37), the presence of endogenous p300 should not affect the YFP experiment.

If autoacetylation is driving the observed behavior they should all be the same if you add excess acetyl-CoA if not you have a folding issue with your construct. – hyperacetylation prevents the formation of condensates, and we observe this trend in all assays.

Given the multiple issues with antibodies it is necessary to at least divide the signal from the PTM AB by a H3 antibody. Or better show it by MS/MS. – we have carefully verified these H3 antibodies in our previous study (ref. 25).

Reviewer 4, Comment 1: *In terms of the model (Fig. 3e), it is not clear why the authors propose that in the hyperacetylated state, the AIL and BD form only “in-cis” and not “in-trans” interactions. Is this based on known structural constraints? Or are they making a local concentration-based argument?*

Author’s response: we have shown by SAXS experiment (Fig. 5) that hyperacetylated p300 remains monomeric, implying that the interaction BD-KacAIL *in trans* is less favorable.

Reviewer 4, Comment 2: *The figures with droplets seem not very sharp. For some of the key experiments (eg. Figs. 1f, 4g, 6f), figures showing the droplets more clearly will strengthen the key arguments. Increasing the time before imaging so that the droplets are settled and more of them are in one plane could help.*

Author’s response: we have replaced Figs. 1f and 4g with the clearer images.

Reviewer 4, Comment 3: *The measurements shown of condensate per 50 um x 50 um area appear be the number of droplets and not the size or volume of droplets. The critical concentration, i.e., the lowest concentration at which phase-separation occurs is typically used in the field to quantify the efficacy of phase-separation. Droplet number (as opposed to cumulative droplet sizes), is more complicated to interpret in terms of phase-separation efficacy as droplet number will be affected by how fast droplets fuse, which in turn will vary based on viscosity effects. Therefore, for the same key experiments mentioned above in #2 a measurement of the saturation/critical concentrations will enable better estimation of the effects of the p300 mutants and of nucleosomes.*

Author’s response: as suggested, we have measured the critical concentration, the data are shown in Suppl. Fig. 2d.

Reviewer 4, Comment 4: *For Fig. 3c, a control with Co-A + p300-BRPHZ+ PEG is needed to rule out that the Co-A part of Acetyl -CoA is acting as a hydrotrope (PMID: 28522535).*

Author’s response: this control has been added in Suppl. Fig. 3a.

Reviewer 4, Comments 5 and 6: *For the nucleosome acetylation assay in Fig. 6j, an additional control is needed to test that PEG does not affect reaction on nucleosomes. This would be analogous to the control in Fig. 6g, where the effect of PEG on auto-acetylation is shown to be negligible. Also, it will be helpful if the authors can speculate why SIRT2 treatment does not affect activity on nucleosomes. Based on the model, it could be expected that the AIL would be better at inhibiting activity in the unacetylated state. Is this because nucleosomes being used are saturating concentrations and that their binding displaces the AIL loop?*

For the nucleosome acetylation data shown in Fig. 6j, it’s possible that the inhibitory effect (grey color) is larger than the ~ 3-fold that can be estimated from the 2 min time-point. This is because the reaction in orange is essentially over by the 2 min time point. If the authors take earlier time points than 2 minutes and plot a rate, they may find that the inhibitory effect is larger than 3-fold, and this would increase the biological significance.

Author’s response: we have added this control (Suppl. Fig. 4a) to confirm that PEG does not decrease the reaction rate on nucleosomes.

Indeed, the SIRT2 treatment does not affect activity on nucleosomes because the reaction in orange/blue is essentially over by the 2 min time point. For each time point, the reactions have to be quenched in parallel, and we had to use a single pipette to avoid error. With multiple samples and controls, we physically could not collect earlier time points.

Reviewer 4, Comment 7: *Some clarification is needed to understand why the authors conclude that p300 is specific for H3K27 over H3K9 based on the figures in 7a-d. From these figures it is not easily apparent how the extent of H3K27 vs H3K9 acetylation can be directly compared given that different antibodies are used for each mark.*

Author's response: we have previously determined the HAT specificity of p300 – it acetylates primarily H3K27 and H3K18 (ref. 25). We also showed that p300 can acetylate non-specifically and to a lesser degree other sites, including H3K9.

REVIEWERS' COMMENTS

Reviewer #1 (Remarks to the Author):

The authors had addressed my concerns in this revision. In my opinion, the revised manuscript is significantly improved and now ready for publication.

Reviewer #2 (Remarks to the Author):

The authors have adequately responded to comments. The final section of the results, concerning the co-localization of p300 and H3K27me3, would be more convincing if additional nuclei were measured - to allow readers a better sense of the extent to which p300 and H3K27 co-localization is truly preferential compared to p300 and H3K4.

Reviewer #3 (Remarks to the Author):

If full length p300 is needed and can't be expressed in E. coli then maybe you should look at other expression systems. It seems that you are already using insect cells and a quick search suggests that others are expressing full-length p300 and there is some cryo-xm data on full length.

If you only use a 37 bp DNA of course the droplets will be smaller. You should repeat with 601 as that is an easy control.

Specificity for K27 is not validated. This needs to be fixed or dropped as it isn't critical for the results of the paper. The ref (25) is based on histone peptides which will not substitute for full-length histones there are plenty of examples of this in the literature for p300. I would simply state that p300 can or is responsible for acetylating the critical residue K27 on histone H3

Reviewer #4 (Remarks to the Author):

The authors' revisions are satisfactory and address my main concerns.